# Differential active site requirements for NDM-1 β-lactamase hydrolysis of carbapenem versus penicillin and cephalosporin antibiotics

Zhizeng Sun [1], Liya Hu [2], Banumathi Sankaran[3], B.V.Venkataram Prasad[2] & Timothy Palzkill[1,2]

New Delhi metallo-β-lactamase-1 exhibits a broad substrate profile for hydrolysis of the penicillin, cephalosporin and 'last resort' carbapenems, and thus confers bacterial resistance to nearly all β-lactam antibiotics. Here we address whether the high catalytic efficiency for hydrolysis of these diverse substrates is reflected by similar sequence and structural requirements for catalysis, i.e., whether the same catalytic machinery is used to achieve hydrolysis of each class. Deep sequencing of randomized single codon mutation libraries that were selected for resistance to representative antibiotics reveal stringent sequence requirements for carbapenem versus penicillin or cephalosporin hydrolysis. Further, the residue positions required for hydrolysis of penicillins and cephalosporins are a subset of those required for carbapenem hydrolysis. Thus, while a common core of residues is used for catalysis of all substrates, carbapenem hydrolysis requires an additional set of residues to achieve catalytic efficiency comparable to that for penicillins and cephalosporins.

[1] Department of Pharmacology and Chemical Biology, Baylor College of Medicine, One Baylor Plaza, Houston, TX 77030, USA. [2] Verna Marrs McLean Department of Biochemistry and Molecular Biology, Baylor College of Medicine, One Baylor Plaza, Houston, TX 77030, USA. [3] Department of Molecular Biophysics and Integrated Bioimaging, Berkeley Center for Structural Biology, Lawrence Berkeley National Laboratory, Berkeley, CA 94720, USA. Correspondence and requests for materials should be addressed to T.P. (email: timothyp@bcm.tmc.edu)

β-Lactam antibiotics (penicillins, cephalosporins, monobactams, and carbapenems) represent the most frequently prescribed antimicrobial agents due to their high efficacy and low toxicity[1]. However, their efficacy is threatened by β-lactamases, which inactivate the antibiotics by hydrolyzing the β-lactam ring. Dissemination of β-lactamase-encoding genes contributes significantly to the development of multidrug resistance observed in various bacterial pathogens including carbapenem-resistant *Enterobacteriaceae*[2].

Based on primary amino-acid sequence homology, β-lactamases fall into four classes: A, B, C, and D. Among these, classes A, C, and D are serine hydrolases that employ an active site serine to catalyze β-lactam hydrolysis, whereas class B consists of metallo-β-lactamases (MBLs), which require one or two zinc ions for activity[3]. MBLs are further classified into B1, B2, and B3 subclasses based on sequence conservation and zinc coordination residues (Fig. 1a). Among the MBLs, subclass B1 enzymes such as impenemase (IMP)-, Verona integron-encoded metallo-β-lactamase (VIM-), and New Delhi metallo-β-lactamase (NDM)-type MBLs are the most clinically relevant[4–7]. In addition to their broad-spectrum activity, many of these enzymes are encoded as gene cassettes and reside with other resistance genes within integrons and plasmids that facilitate their rapid dissemination through horizontal gene transfer[7]. In addition, although there has been recent progress in the development of inhibitors[8,9], MBLs are not susceptible to clinically available β-lactamase inhibitors[7]. Therefore, MBLs pose a global threat to public health.

A number of crystal structures of NDM-1 in complex with various hydrolyzed β-lactam substrates have been reported[10–13]. These suggest a catalytic mechanism for di-zinc MBLs in which the substrate binds through interaction of the carbonyl oxygen of β-lactam ring with the side chain of Asn233 and Zn1 and the carboxyl group on the fused β-lactam ring interacts with Zn2 and residue Lys224 (MBL numbering) (Fig. 1b)[14]. A hydroxide ion that is stabilized by Zn1 and Zn2 then attacks the carbonyl carbon of the β-lactam ring leading to cleavage of the C–N amide bond[13,15]. Upon opening of the β-lactam ring, an anionic intermediate is generated and the newly formed carboxylate interacts with Zn1 and the amide nitrogen and carboxylate from the fused ring interact with Zn2[13,15,16]. The anionic intermediate is subsequently protonated in the rate-limiting step and product is released[16].

Other than the ligands for the active site Zn ions, the active site of NDM-1 is composed largely of hydrophobic residues. Lys224 and Asn233 are the only residues that make polar interactions with the antibiotic substrate. Lys224 binds the C3/C4 carboxylate of β-lactam substrates and Asn233 interacts with C6 carbonyl oxygen and, after hydrolysis, the newly formed carboxylate of the product[11]. These interactions are proposed to facilitate binding and orienting the substrate and, based on hybrid quantum chemical/molecular mechanical (QMM-MM) simulations, retain contact with intermediates throughout the catalytic cycle[17].

Studies of enzyme kinetic parameters of purified NDM-1 β-lactamase reveal a broad substrate profile with $k_{cat}/K_M$ values of $\sim1–5\times10^6\,M^{-1}\,s^{-1}$ for penicillins and carbapenems and $\sim1\times10^7\,M^{-1}\,s^{-1}$ for cephalosporins, whereas monobactams are not hydrolyzed[18]. A study on $k_{cat}/K_M$ values for thousands of enzymes suggests an average $k_{cat}/K_M$ value of $\sim10^5\,M^{-1}\,s^{-1}$[19]. Therefore, NDM-1 not only catalyzes the hydrolysis of a wide range of β-lactams—it does so with high efficiency. This occurs despite considerable differences in antibiotic structure beyond the

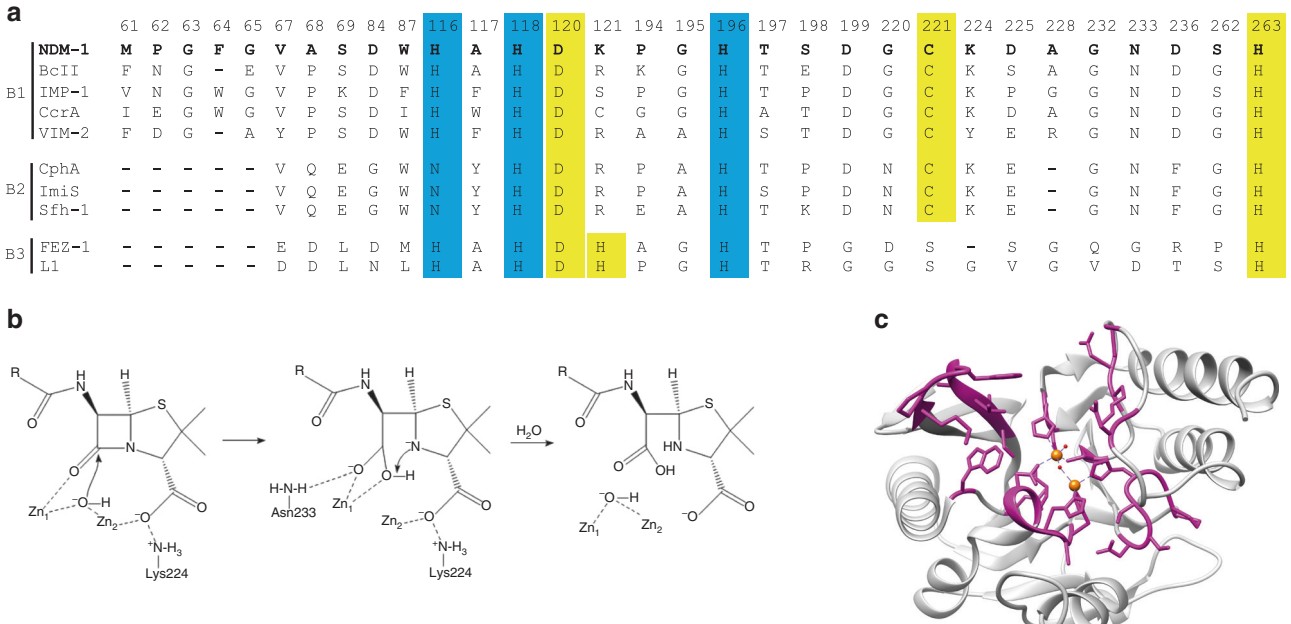

**Fig. 1** Active site residues of metallo-β-lactamases. **a** Sequence alignment of representative metallo-β-lactamases from subclasses B1, B2, and B3. The amino acids highlighted in blue and yellow indicate the histidine and cysteine zinc binding site residues, respectively. NDM-1 residues are in bold. **b** Schematic representation of β-lactam substrate binding (penicillin), anionic intermediate stabilization, and product release in the active site of di-zinc metallo-β-lactamase NDM-1. The substrate binds to the active site through interaction of the carbonyl oxygen of the β-lactam ring with Zn1 and the carboxyl group on the fused ring with Zn2 and residue Lys224. A hydroxide ion stabilized by Zn1 and Zn2 attacks the carbonyl carbon of the β-lactam ring, leading to the formation of a carboxylate group and a nitrogen anion. The former is coordinated by Zn1 and the side chain of the conserved Asn233. The latter is stabilized by Zn2 and protonated coincident with or after C–N bond cleavage. The proton donor for the anionic nitrogen is shown as the newly formed carboxylate, however, the proton has also been proposed to be donated by a water. **c** Diagram of the NDM-1 β-lactamase structure highlighting active site residues for which random mutant libraries were created (magenta). The zinc atoms are represented as orange spheres. The figure was rendered with using coordinates from the Protein Data Bank accession code 3SPU[18]

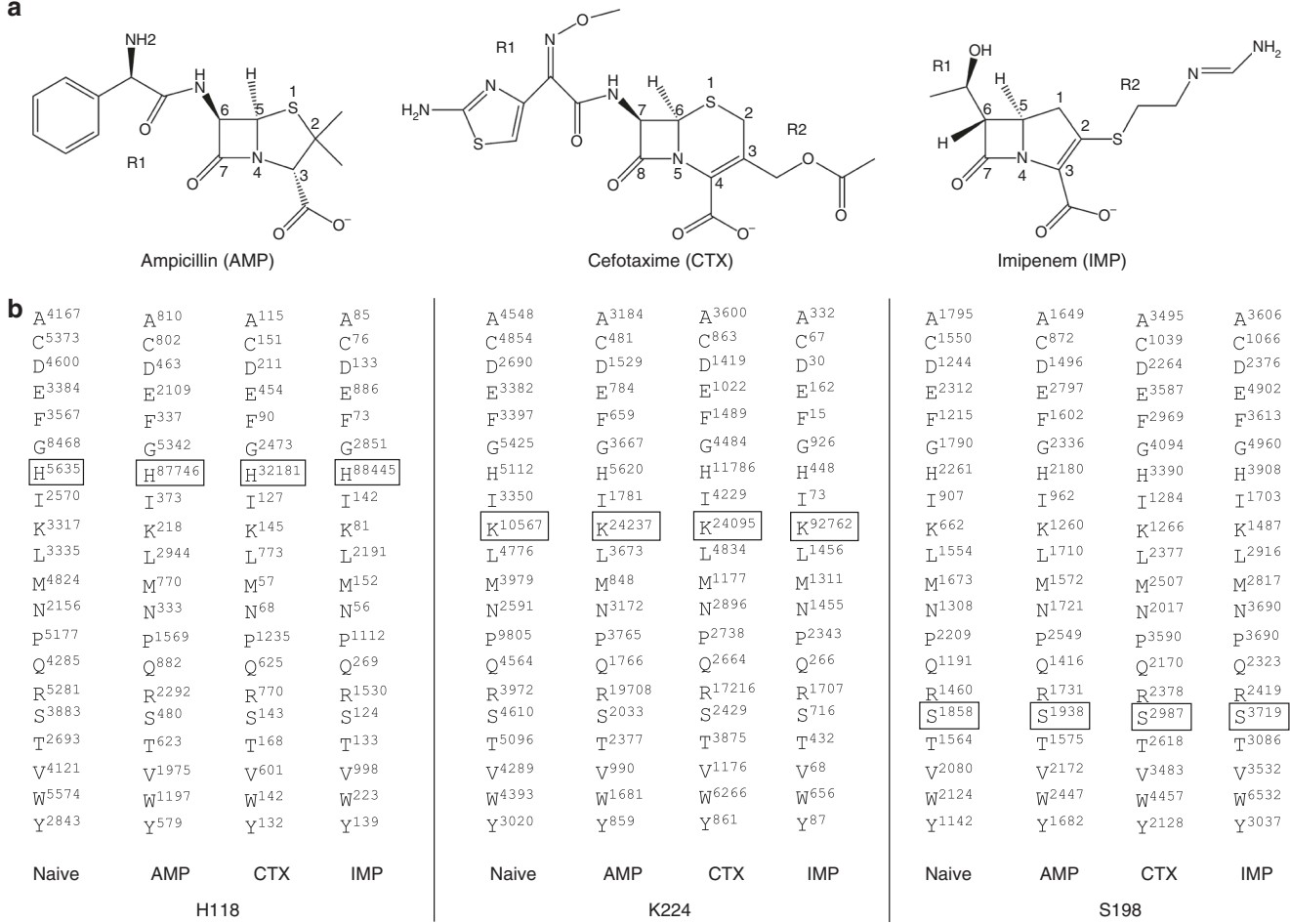

**Fig. 2** Antibiotic structures and representative DNA sequencing results. **a** Structures of ampicillin, cefotaxime, and imipenem. The position of the R1 group is indicated on the structure of ampicillin and the R1 and R2 groups are indicated on the structures of cefotaxime and imipenem. **b** Representative DNA sequencing results of clones from naive and antibiotic-selected NDM-1 libraries. A total of 124 experiments from 31 libraries were sequenced and the results from three representative libraries are shown. The number of times the wild-type amino acid occurred in the sequenced library is boxed in each experiment. AMP ampicillin, CTX cefotaxime, IMP imipenem

common β-lactam ring for penicillins, cephalosporins, and carbapenems (Fig. 2).

An interesting question is whether the high level of catalytic efficiency observed for these varied substrates is achieved via contributions from the same constellation of active site residues. To address this, a codon randomization and functional selection approach was used. This involved creating single codon randomized libraries for residue positions in and near the active site of NDM-1 and selecting for mutants from each library that support growth of *Escherichia coli* (*E. coli*) in the presence of a representative penicillin (ampicillin (AMP)), cephalosporin (cefotaxime (CTX)), and carbapenem (IMP) antibiotic (Figs. 1c, 2a). Deep sequencing of functional mutants from each library yielded the frequency of occurrence of each amino acid at each randomized position after antibiotic selection. A comparison of the sequencing results from each antibiotic selection revealed that wild-type amino-acid residues predominate at eight residue positions in all antibiotic selection experiments, implying that there is a core set of residues that contribute to the hydrolysis of all classes of β-lactams. However, the predominance of wild-type amino-acid residues at several other residue positions was dependent on the antibiotic used for selection, indicating context-dependent sequence requirements. The majority of these positions showed IMP-specific effects with substitutions tolerated for AMP and

CTX hydrolysis but not for IMP activity, suggesting more stringent requirements for carbapenem hydrolysis.

The stringent sequence requirements for carbapenem hydrolysis were illustrated by the construction of a triple mutant (K224R/G232A/N233Q) based on the sequencing results that hydrolyzes AMP at wild-type levels while essentially losing the ability to hydrolyze IMP, thereby disconnecting IMP and AMP catalysis. The X-ray structure of the triple mutant revealed that the mutations caused local conformational changes of an active site loop leading to the loss of contacts between NDM-1 and β-lactam substrates that are critical for IMP catalysis but not for AMP catalysis. Taken together, the results indicate that the overall sequence requirements in the active site vary between the antibiotics with more stringent sequence requirements for the hydrolysis of a representative carbapenem compared with that for a penicillin and cephalosporin, despite the fact that all are catalyzed at similar levels by the wild-type enzyme. Thus, hydrolysis of IMP is a difficult task, suggesting that carbapenem hydrolysis requires the contribution of significantly more residues near the active site of NDM-1 β-lactamase.

## Results

**Deep sequencing of NDM-1 randomized codon libraries.** Crystal structures of NDM-1 β-lactamase in complex with

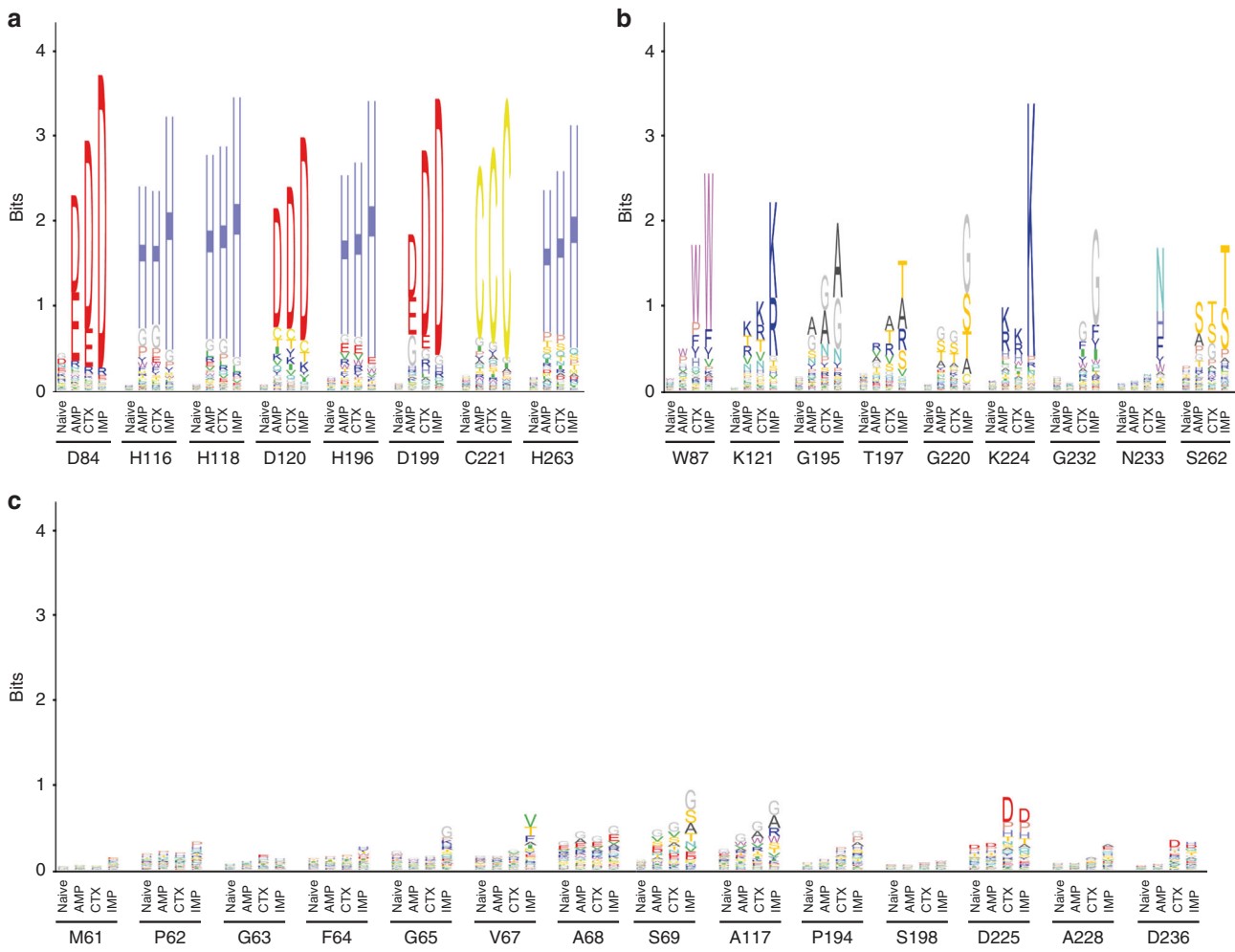

**Fig. 3** Sequence logos for essential, substrate-specific, and non-essential residue classes. **a** Essential residue class. For this class, the sequences obtained from the antibiotic-selected library are dominated by the wild-type residues and sometimes physicochemically similar residues. **b** Substrate-specific residue class. For this class, the distribution of residue types observed is dependent on the antibiotic used for selection. For example, a broad distribution of amino-acid types is observed among the sequences from ampicillin-selected libraries, whereas a narrower range of amino-acid types is observed among sequences from imipenem-selected libraries. **c** Non-essential residue class. For this class, a wide distribution of amino-acid types is observed after selection of the libraries with each of the β-lactam antibiotics, suggesting relaxed sequence requirements at these positions

hydrolyzed penicillin[11,12], cephalosporin[10], and carbapenem[11,13] antibiotics have localized active site residues involved in substrate binding and catalysis. In order to examine the importance of the active site residues for hydrolysis of various β-lactam antibiotics, randomized single codon libraries targeting 31 residues in and near the active site of NDM-1 were constructed (Fig. 1c). Functional mutants were selected from each library based on the ability to confer growth on agar plates containing AMP, CTX, or IMP (Fig. 2a). Appropriate concentrations of antibiotics were used to select for phenotypically wild-type levels of β-lactam resistance. Resistant clones were pooled and regions of the $bla_{NDM-1}$ gene containing randomized codons were PCR amplified with primers with unique barcode sequences for each experiment. The barcode-tagged amplicons were then pooled and analyzed using next-generation sequencing. A naive library control that was not subjected to antibiotic selection was included in each experiment. The PCR amplicons that were pooled represent 124 experiments—each of the 31 random libraries selected with three antibiotics and an unselected control (31 libraries × 4).

Computational processing of the sequencing data yielded 3.1 × $10^7$ total sequences for the 124 experiments (2.5 × $10^5$ sequences for each experiment on average) from which the number of

occurrences of each amino-acid type in each experiment was determined (Methods) (Fig. 2b, Supplementary Data 1). For the naive library, each amino-acid type occurred at a comparable frequency at each position except for Gly65, Gly195, and Gly232, in which glycine occurred at a significantly higher frequency than other amino acids (Supplementary Data 1). This bias may occur during PCR or Illumina sequencing as it was not observed by Sanger sequencing of the NDM-1 gene from individual colonies.

To visualize sequence conservation among functional mutants from the antibiotic-selected libraries, sequence logos were created based on the deep sequencing results (Methods) (Fig. 3). The NDM-1 active site residues can be placed into three groups based on the frequencies of substitutions revealed by the sequencing results (Figs. 3, 4). The first group includes residue positions where the wild-type amino acid predominates among the functional mutants selected on each of the antibiotics (Fig. 3a, Supplementary Data 1). This group represents a core set of residues that contribute strongly to the hydrolysis of all antibiotics tested. These positions include the $Zn^{2+}$-chelating residues (His116, His118, Asp120, His196, Cys221, and His263) indicating that both $Zn^{2+}$ sites are required for hydrolysis of all of the β-lactams tested and that substitution of alternative residues

that could bind $Zn^{2+}$ at these positions results in decreased enzyme activity (Fig. 4). In addition to the zinc-chelating residues, the wild-type aspartate at positions 84 and 199 predominated among functional mutants selected on each of the antibiotics. Asp84 is part of an extensive hydrogen-bonding network that forms the shell around the Zn2 site and, presumably, substitutions disrupt this network (Fig. 4). Asp199 is part of a hydrogen-bonding network that links loop L6, L9 and helix 5 near Zn1 and, similar to Asp84, substitutions may disrupt this network.

The second group of residues exhibited context-dependent sequence requirements (Fig. 3b). For these positions, the distribution of amino acids found among functional clones depends on the antibiotic used for selection. Residue positions in this group include Trp87, Lys121, Gly195, Thr197, Gly220, Lys224, Gly232, Asn233, and Ser262 (Figs. 3b, 4). For this group, substitutions decrease NDM-1 activity towards IMP but not AMP or CTX. Thus, more residues in the active site contribute to IMP hydrolysis compared with AMP and CTX hydrolysis. This result indicates that the sequence requirements for NDM function are different for different substrates despite the wild-type enzyme

displaying similar catalytic efficiency for hydrolysis of all the antibiotics.

The final and largest group of residue positions displayed a wide range of amino acids among functional clones with no dominant residue type (Fig. 3c). This group included residues Met61through Ser69, Ala117, Pro194, Ser198, Asp225, Ala228, and Asp236 (Figs. 3c, 4; Supplementary Data 1). These residue positions do not have precise sequence requirements in order to carry out their role in enzyme function. Note, however, that although several residue types are consistent with function at these positions, not all substitutions are allowed. For example, cysteine occurs at a low frequency for many of these positions (Supplementary Data 1).

It is qualitatively apparent from the sequence logos in Figs. 3a, b that the sequence requirements for IMP hydrolysis are more stringent than for AMP and CTX hydrolysis. In order to obtain a quantitative assessment of sequence variability after selection, the effective number of amino-acid substitutions ($k^*$) was determined at each position in the naive and antibiotic-selected libraries[20,21]. The value of $k^*$ was calculated from the substitution frequencies as described in Methods[21]. A $k^*$ value of 1 indicates that only one amino-acid type is found in the library for a specific position, whereas a $k^*$ value of 20 indicates that all 20 amino acids occur at equal frequency, i.e., there is maximal diversity. $k^*$ values were close to 20 for the naive libraries as expected for randomized positions (Fig. 5).

The $k^*$ values for the residue positions after IMP selection are lower than the corresponding values for the AMP and CTX-selected clones for nearly every residue position (Fig. 5). However, the difference is particularly apparent for the residues that exhibit context-dependent sequence requirements including residues Trp87, Lys121, Thr197, Gly220, Lys224, Gly232, Asn233, and Ser262 (Fig. 5). For all of these positions, the effective number of substitutions is highest for AMP, followed by CTX, and with IMP-selected mutants having lower values. These results indicate that more sequence diversity is tolerated at these residue positions among the AMP- and CTX-selected mutants compared with those among the IMP-selected mutants.

**Validation of deep sequencing results**. Deep sequencing of antibiotic-selected mutation libraries of NDM-1 identified three classes of residue positions based on tolerance to amino-acid substitutions (Fig. 3). In order to validate the results using a different method, a number of mutants from each class were further characterized by determining antibiotic minimum inhibitory concentrations (MICs) (Methods).

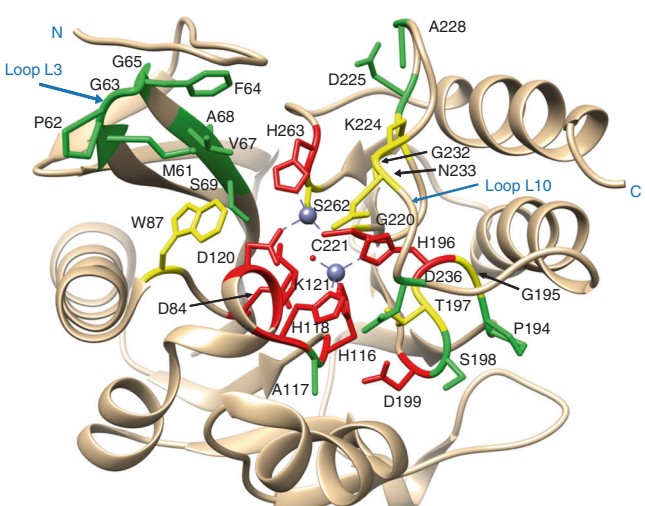

**Fig. 4** Diagram of the NDM-1-β-lactamase structure showing the location of essential, substrate-specific, and non-essential residues. Essential, substrate-specific, and non-essential residues are labeled red, yellow, and green, respectively, whereas the zinc atoms are represented as gray spheres. The figure was rendered using coordinates from the Protein Data Bank accession code 3Q6X[6]

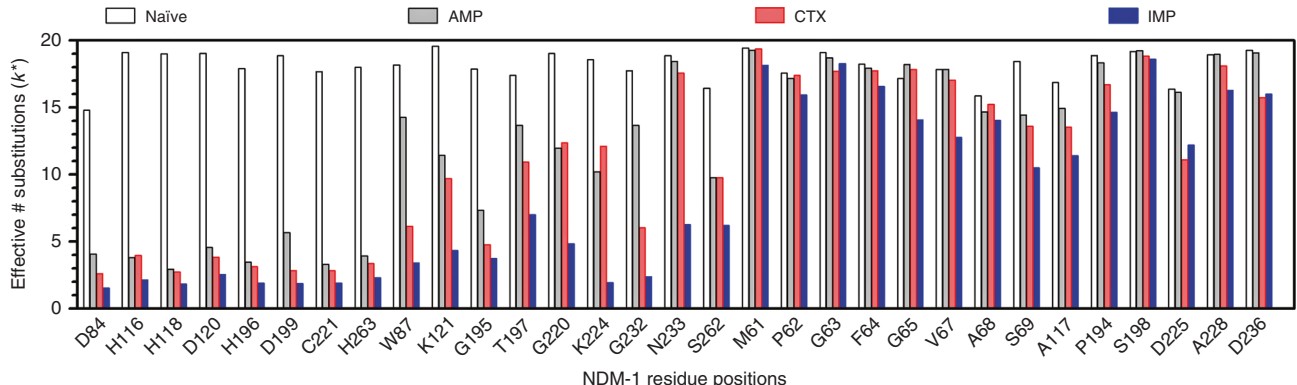

**Fig. 5** Bar chart of the effective number of substitutions ($k^*$) for randomized positions. $k^*$ values for the naive libraries (white) and libraries after selection with ampicillin (gray), cefotaxime (red), and imipenem (blue) are shown

**Table. 1 Antibiotic resistance levels of wild-type and mutants**

| Expressed protein | MIC (µg ml⁻¹) | | |
|---|---|---|---|
| | Ampicillin | Cefotaxime | Imipenem |
| None | 4 | 0.25 | 0.25 |
| NDM-1 WT | 4096 | 256 | 16 |
| F64H | 4096 | 256 | 16 |
| D84E | 256 | 16 | 1 |
| W87P | 2048 | 32 | 2 |
| K121R | 512 | 64 | 4 |
| K121T | 512 | 64 | 2 |
| G195A | 4096 | 256 | 8 |
| D199E | 1024 | 64 | 2 |
| K224H | 256 | 128 | 0.5 |
| K224R | 2048 | 256 | 4 |
| D225E | 4096 | 128 | 16 |
| G232A | 2048 | 64 | 1 |
| N233H | 4096 | 256 | 8 |
| N233Q | 2048 | 256 | 2 |

*MIC* minimum inhibitory concentration

Consistent with the deep sequencing results, the essential role of zinc-chelating residues for all substrates has been well characterized previously by site-directed mutagenesis studies[22]. The Asp84 and Asp199 residues were also identified as essential for all substrates by deep sequencing and, consistent with these findings, glutamate substitutions at these positions markedly decreased resistance toward the three β-lactam antibiotics as indicated by reduced MICs (Table 1). In addition, amino-acid substitutions at context-dependent residue positions Trp87, Lys224, Gly232, and Asn233 confirmed that the effect of mutations at these positions on the antibiotic resistance function of NDM-1 depends on the antibiotic used for testing (Table 1). Finally, substitutions at residue positions that can be freely substituted and retain function for all antibiotics including the F64H and D225E mutants, did not exhibit altered resistance levels against any tested antibiotics, consistent with the deep sequencing results (Table 1).

In order to further test the deep sequencing results, multiple clones were randomly picked from naive libraries for two residue positions for each of the three classes of positions from Fig. 3. Clones from the naive libraries for residue positions Asp84 and His263 were picked to represent the essential class, Lys224 and Gly232 for the context-dependent class, and Val67 and Asp236 for the non-essential class. The amino-acid substitution for each of these clones was determined by Sanger sequencing and their antibiotic resistance levels were quantified by MIC values. The relative fitness $f_u^a$ of each mutant compared with wild-type was calculated from MIC values of *E. coli* cells containing wild-type or mutant NDM-1 as described in Methods. In addition, the relative fitness $F_u^a$ of each mutant versus wild-type was determined from the deep sequencing results based on the frequency of occurrence of each mutant allele versus the occurrence of the wild-type allele in naive and antibiotic selection experiments, as described in Methods. If the frequencies obtained from deep sequencing are a reflection of the in vivo activity of the mutants as measured by MIC, there should be a correlation between the relative fitness based on MIC values versus that determined from the frequency of occurrence from sequencing. Plotting the $f_u^a$ values versus the corresponding $F_u^a$ values for each antibiotic selection experiment tested this relationship. As shown in Supplementary Fig. 1, linear regression analysis of the plots indicates a significant positive

correlation between $f_u^a$ and $F_u^a$ for each antibiotic experiment. Taken together, the MIC results for mutants from each selection and those chosen from naive libraries are consistent with the ability of deep sequencing experiments to sort mutants based on levels of resistance for each antibiotic tested.

**Effect of substitutions on catalysis and protein expression**. The antibiotic resistance levels provided by NDM-1 and its mutants reflect their ability to hydrolyze the β-lactam antibiotics and the steady-state level of the protein expressed in the periplasm. To assess the effect of amino-acid substitutions on enzyme activity of NDM-1 β-lactamase, the wild-type and mutant enzymes from each of the three classes of positions from Fig. 3 were purified and kinetic parameters $k_{cat}$, $K_M$, and $k_{cat}/K_M$ were determined (Table 2). In addition, steady-state protein expression levels were measured by western blot using an antibody that recognizes the StrepII tag present at the C-terminus of the wild-type and mutant enzymes (Supplementary Fig. 2).

Because the zinc ligand residues in subclass B1 enzymes have been extensively studied and substitutions at these positions are known to decrease enzyme activity[3], we focused on the residues from the essential class that are not zinc ligands. The Asp84 and Asp199 residues are essential for resistance to all tested antibiotics based on sequencing results. Although not in position to contact zinc or the β-lactam substrate, they make key hydrogen bonds that link sections of the active site together. Therefore, they are hypothesized to be important for NDM-1 structure and stability. Enzyme kinetics results revealed glutamate substitutions at these positions did not significantly affect catalytic efficiency ($k_{cat}/K_M$) of NDM-1 β-lactamase against any tested antibiotic (Table 2) but dramatically decreased cellular expression levels of the enzyme (Supplementary Fig. 2). This indicates that residues Asp84 and Asp199 are not essential for β-lactamase activity but are important for the stable expression of NDM-1.

Mutations at context-dependent residue positions Trp87, Lys224, Gly195, Gly232, and Asn233 displayed a substrate-dependent effect on NDM-1 enzyme activity (Table 2). In general, substitutions at these residues decreased the $k_{cat}/K_M$ value for IMP hydrolysis more than that observed for AMP and CTX hydrolysis. For example, when compared with wild-type NDM-1, the K224R mutant enzyme displayed a fourfold lower $k_{cat}/K_M$ for IMP hydrolysis but only a twofold lower AMP hydrolysis and wild-type levels of CTX hydrolysis. In addition, the N233Q enzyme displayed sevenfold lower IMP hydrolysis but twofold lower and twofold higher levels of hydrolysis of AMP and CTX, respectively. Further, AMP and CTX hydrolysis by NDM-1 was increased by the N233H mutation, whereas imipenemase activity was slightly decreased by the same mutation. Therefore, the enzyme kinetics results are consistent with the sequencing results and reveal the efficient hydrolysis of IMP has more stringent sequence requirements at these residue positions than hydrolysis of AMP or CTX by NDM-1 β-lactamase.

Finally, as expected based on the deep sequencing results, amino-acid substitutions at non-essential residue positions Phe64 and Asp225 did not significantly compromise the β-lactamase activity of NDM-1 (Table 2), confirming these positions can be substituted and the enzyme retains high-level function. In addition, western blot results show the F64H and D225E mutants are expressed at similar levels as wild-type NDM-1, further confirming these positions can be substituted without affecting structure and function (Supplementary Fig. 2).

**Combining substitutions greatly reduces carbapenem catalysis**. Deep sequencing of antibiotic-selected mutation libraries demonstrated that the active site of NDM-1 β-lactamase has more

**Table. 2 Enzyme kinetic parameters for NDM-1 and mutants**

| Proteins | Kinetic parameters | Substrate[a] | | |
|---|---|---|---|---|
| | | Ampicillin | Cefotaxime | Imipenem |
| NDM-1 | $K_M$ (μM) | 143 ± 3 | 6 ± 1 | 65 ±± 10 |
| | $k_{cat}$ (s⁻¹) | 550 ± 15 | 85 ± 16 | 238 ± 16 |
| | $k_{cat}/K_M$ (μM⁻¹ s⁻¹) | 3.84 ± 0.18 | 13.79 ± 1.49 | 3.71 ± 0.36 |
| F64H | $K_M$ (μM) | 188 ± 31 | 6 ± 0.1 | 164 ± 19 |
| | $k_{cat}$ (s⁻¹) | 793 ± 24 | 83 ± 2 | 409 ± 15 |
| | $k_{cat}/K_M$ (μM⁻¹ s⁻¹) | 4.27 ± 0.57 | 14.80 ± 0.62 | 2.50 ± 0.20 |
| D84E | $K_M$ (μM) | 82 ± 6 | 16 ± 0.2 | 70 ± 5 |
| | $k_{cat}$ (s⁻¹) | 597 ± 17 | 113 ± 3 | 153 ± 3 |
| | $k_{cat}/K_M$ (μM⁻¹ s⁻¹) | 7.29 ± 0.29 | 6.77 ± 0.23 | 2.20 ± 0.12 |
| W87P | $K_M$ (μM) | 518 ± 36 | 6 ± 1 | 295 ± 54 |
| | $k_{cat}$ (s⁻¹) | 1823 ± 8 | 46 ± 3 | 351 ± 49 |
| | $k_{cat}/K_M$ (μM⁻¹ s⁻¹) | 3.53 ± 0.23 | 8.50 ± 1.84 | 1.20 ± 0.05 |
| K121R | $K_M$ (μM) | 61 ± 3 | 4 ± 0.4 | 31 ± 9 |
| | $k_{cat}$ (s⁻¹) | 26 ± 0.35 | 13 ± 0.3 | 36 ± 2 |
| | $k_{cat}/K_M$ (μM⁻¹ s⁻¹) | 0.43 ± 0.03 | 3.69 ± 0.31 | 1.20 ± 0.28 |
| K121T | $K_M$ (μM) | 231 ± 0.71 | 10 ± 2 | 117 ± 13 |
| | $k_{cat}$ (s⁻¹) | 246 ± 8 | 50 ± 0.3 | 50 ± 2 |
| | $k_{cat}/K_M$ (μM⁻¹ s⁻¹) | 1.06 ± 0.03 | 5.13 ± 0.79 | 0.43 ± 0.03 |
| G195A | $K_M$ (μM) | 57 ± 14 | 4 ± 0.5 | 52 ± 15 |
| | $k_{cat}$ (s⁻¹) | 248 ± 24 | 32 ± 0.04 | 94 ± 11 |
| | $k_{cat}/K_M$ (μM⁻¹ s⁻¹) | 4.43 ± 0.67 | 7.74 ± 0.85 | 1.84 ± 0.31 |
| D199E | $K_M$ (μM) | 11 ± 2 | 6 ± 0.4 | 6 ± 0.8 |
| | $k_{cat}$ (s⁻¹) | 23 ± 2 | 48 ± 0.7 | 17 ± 1 |
| | $k_{cat}/K_M$ (μM⁻¹ s⁻¹) | 2.10 ± 0.12 | 7.65 ± 0.40 | 2.81 ± 0.18 |
| K224H | $K_M$ (μM) | 906 ± 26 | 26 ± 0.6 | ND |
| | $k_{cat}$ (s⁻¹) | 330 ± 6 | 111 ± 0.3 | ND |
| | $k_{cat}/K_M$ (μM⁻¹ s⁻¹) | 0.36 ± 0.004 | 4.24 ± 0.076 | 0.086 ± 0.002 |
| K224R | $K_M$ (μM) | 179 ± 17 | 3 ± 0.2 | 69 ± 2 |
| | $k_{cat}$ (s⁻¹) | 294 ± 22 | 39 ± 2 | 62 ± 0.2 |
| | $k_{cat}/K_M$ (μM⁻¹ s⁻¹) | 1.65 ± 0.04 | 12.81 ± 0.28 | 0.90 ± 0.03 |
| D225E | $K_M$ (μM) | 41 ± 9 | 4 ± 0.1 | 50 ± 8 |
| | $k_{cat}$ (s⁻¹) | 102 ± 0.8 | 38 ± 0.1 | 117 ± 8 |
| | $k_{cat}/K_M$ (μM⁻¹ s⁻¹) | 2.54 ± 0.56 | 9.78 ± 0.17 | 2.37 ± 0.24 |
| G232A | $K_M$ (μM) | 178 ± 40 | 24 ± 2 | ND |
| | $k_{cat}$ (s⁻¹) | 417 ± 24 | 90 ± 1 | ND |
| | $k_{cat}/K_M$ (μM⁻¹ s⁻¹) | 2.39 ± 0.41 | 3.80 ± 0.25 | 0.63 ± 0.008 |
| N233H | $K_M$ (μM) | 56 ± 3 | 3 ± 0.5 | 134 ± 43 |
| | $k_{cat}$ (s⁻¹) | 591 ± 6 | 55 ± 3 | 286 ± 30 |
| | $k_{cat}/K_M$ (μM⁻¹ s⁻¹) | 10.58 ± 0.61 | 20.42 ± 2.83 | 2.22 ± 0.49 |
| N233Q | $K_M$ (μM) | 160 ± 13 | 1.1 ± 0.3 | 565 ± 84 |
| | $k_{cat}$ (s⁻¹) | 279 ± 23 | 25 ± 1 | 288 ± 31 |
| | $k_{cat}/K_M$ (μM⁻¹ s⁻¹) | 1.74 ± 0.01 | 22.95 ± 6.85 | 0.51 ± 0.02 |

ND not determined
[a]Data are mean and standard deviations of at least two independent experiments

stringent sequence requirements for IMP hydrolysis compared to the hydrolysis of AMP and CTX. This was particularly apparent for the class of residue positions that exhibit context-dependent sequence requirements including Trp87, Gly195, Thr197, Lys224, Gly232, Asn233, and Ser262. These observations predict that it

should be possible to construct an NDM-1 variant that retains the ability to hydrolyze AMP and CTX while losing the ability to hydrolyze IMP. This idea was tested by combining substitutions at positions Lys224, Gly232, and Asn233 that negatively affect IMP hydrolysis but do not affect AMP or CTX hydrolysis. As noted above, Lys224 and Asn233 are the only NDM residues that make salt bridge or hydrogen-bonding interactions with β-lactam substrates and Gly232 is positioned to make hydrophobic contacts with substrate.

The mutations chosen for this experiment were K224R, G232A, and N233Q, which significantly decrease IMP hydrolysis while having modest effects on AMP and CTX hydrolysis (Table 2). Each of the double mutant combinations, including K224R/G232A, G232A/N233Q, and K224R/N233Q, resulted in significantly decreased IMP hydrolysis compared with either of the single mutant parents and wild-type NDM-1 with $k_{cat}/K_M$ values decreased 100-fold for K224R/G232A and G232A/N233Q and 20-fold for K224R/N233Q compared with wild-type (Table 3). In contrast, each of the double mutants retained high levels of AMP hydrolysis activity (Table 3). Finally, the double mutant K224R/N233Q still retained high levels of CTX hydrolysis, but the K224R/G232A and G232A/N233Q mutants exhibited greatly reduced catalytic efficiency ($k_{cat}/K_M$) for CTX.

Further combining the substitutions resulted in a triple mutant, K224R/G232A/N233Q, which is as active as the wild-type enzyme in hydrolyzing AMP but exhibits a 600-fold reduction in $k_{cat}/K_M$ for IMP hydrolysis. The loss in IMP hydrolysis activity is at least in part due to a greatly increased $K_M$ value, which was too high to be accurately measured. In contrast, the triple mutant has a threefold lower $K_M$ for AMP hydrolysis than wild-type NDM-1. The triple mutant exhibits a similar pH profile as wild-type for AMP hydrolysis, although with a narrower pH optimum, suggesting it operates via a similar mechanism (Supplementary Fig. 3). The triple mutant also exhibits a 50-fold reduction in $k_{cat}/K_M$ for CTX hydrolysis, due to a 10-fold increase in $K_M$ and 5-fold decrease in $k_{cat}$. Comparison of the K224R/N233Q mutant with the triple mutant indicates the majority of the decreased activity for CTX is due to the G232A substitution (Table 3). The properties of the triple mutant are consistent with the view that efficient IMP hydrolysis requires conservation of a larger number of residues than are required for AMP hydrolysis. To investigate whether this finding can be generalized to β-lactams in the same class, the triple mutant K224R/G232A/N233Q was tested for hydrolysis of another penicillin and carbapenem, i.e., benzylpenicillin and meropenem. As shown in Supplementary Table 1, compared with wild-type NDM-1 enzyme, the triple mutant displays a 25-fold lower $k_{cat}/K_M$ value for hydrolyzing benzylpenicillin but a 250-fold lower $k_{cat}/K_M$ value for hydrolyzing meropenem. Therefore, the result is consistent with the idea that more extensive amino-acid sequence information is required in the active site of NDM-1 for carbapenem hydrolysis compared with other β-lactam antibiotics.

**Structural basis for activity of NDM-1 triple mutant**. Although wild-type NDM-1 enzyme displays comparable activity against AMP, CTX, and IMP[18], the triple mutant NDM-1 K224R/G232A/N233Q preferentially hydrolyzes AMP (Table 3). In order to understand the basis for its altered substrate specificity, the structure of the triple mutant was determined by X-ray crystallography to 1.75 Å resolution with four molecules in the asymmetric unit (Supplementary Figs. 4, 5). In each chain of NDM-1 K224R/G232A/N233Q, there are zinc ions coordinated to the histidine site and cysteine site. A water molecule or hydroxide ion was found between the two zinc ions in each chain. An overlay of the four chains revealed slight structural variations among them

**Table. 3 Enzyme kinetic parameters for NDM-1 single and combinatorial mutants**

| Proteins | Kinetic parameters | Substrate[a] | | |
|---|---|---|---|---|
| | | Ampicillin | Cefotaxime | Imipenem |
| NDM-1 WT[b] | $K_M$ (µM) | 143 ± 3 | 6 ± 1 | 65 ± 10 |
| | $k_{cat}$ (s$^{-1}$) | 550 ± 15 | 85 ± 16 | 238 ± 16 |
| | $k_{cat}/K_M$ (µM$^{-1}$s$^{-1}$) | 3.84 ± 0.18 | 13.79 ± 1.49 | 3.71 ± 0.36 |
| K224R[b] | $K_M$ (µM) | 179 ± 17 | 3 ± 0.2 | 69 ± 2 |
| | $k_{cat}$ (s$^{-1}$) | 294 ± 22 | 39 ± 2 | 62 ± 0.2 |
| | $k_{cat}/K_M$ (µM$^{-1}$s$^{-1}$) | 1.65 ± 0.04 | 12.81 ± 0.28 | 0.90 ± 0.03 |
| G232A[b] | $K_M$ (µM) | 178 ± 40 | 24 ± 2 | ND |
| | $k_{cat}$ (s$^{-1}$) | 417 ± 24 | 90 ± 1 | ND |
| | $k_{cat}/K_M$ (µM$^{-1}$s$^{-1}$) | 2.39 ± 0.41 | 3.80 ± 0.25 | 0.63 ± 0.008 |
| N233Q[b] | $K_M$ (µM) | 160 ± 13 | 1.1 ± 0.3 | 565 ± 84 |
| | $k_{cat}$ (s$^{-1}$) | 279 ± 23 | 25 ± 1 | 288 ± 31 |
| | $k_{cat}/K_M$ (µM$^{-1}$s$^{-1}$) | 1.74 ± 0.01 | 22.95 ± 6.85 | 0.51 ± 0.02 |
| K224R/G232A | $K_M$ (µM) | 93 ± 11 | 78 ± 18 | ND |
| | $k_{cat}$ (s$^{-1}$) | 108 ± 11 | 33 ± 6 | ND |
| | $k_{cat}/K_M$ (µM$^{-1}$s$^{-1}$) | 1.16 ± 0.018 | 0.43 ± 0.02 | 0.038 ± 0.0028 |
| G232A/N233Q | $K_M$ (µM) | 72 ± 11 | 70 ± 3 | ND |
| | $k_{cat}$ (s$^{-1}$) | 263 ± 11 | 29 ± 1 | ND |
| | $k_{cat}/K_M$ (µM$^{-1}$s$^{-1}$) | 3.68 ± 0.43 | 0.42 ± 0.012 | 0.039 ± 0.0016 |
| K224R/N233Q | $K_M$ (µM) | 43 ± 3 | 5.37 ± 0.43 | ND |
| | $k_{cat}$ (s$^{-1}$) | 263 ± 11 | 22 ± 0.20 | ND |
| | $k_{cat}/K_M$ (µM$^{-1}$s$^{-1}$) | 6.08 ± 0.20 | 4.02 ± 0.28 | 0.17 ± 0.0013 |
| K224R/G232A/N233Q | $K_M$ (µM) | 51 ± 6 | 78 ± 5 | ND |
| | $k_{cat}$ (s$^{-1}$) | 180 ± 5 | 18 ± 0.8 | ND |
| | $k_{cat}/K_M$ (µM$^{-1}$s$^{-1}$) | 3.52 ± 0.34 | 0.24 ± 0.005 | 0.0064 ± 0.00012 |

*ND* not determined
[a]Data are mean and standard deviations of at least two independent experiments
[b]Values are from Table 1 for comparison between wild-type, single, and combinatorial mutations at residues Lys224, Gly232, and Asn233

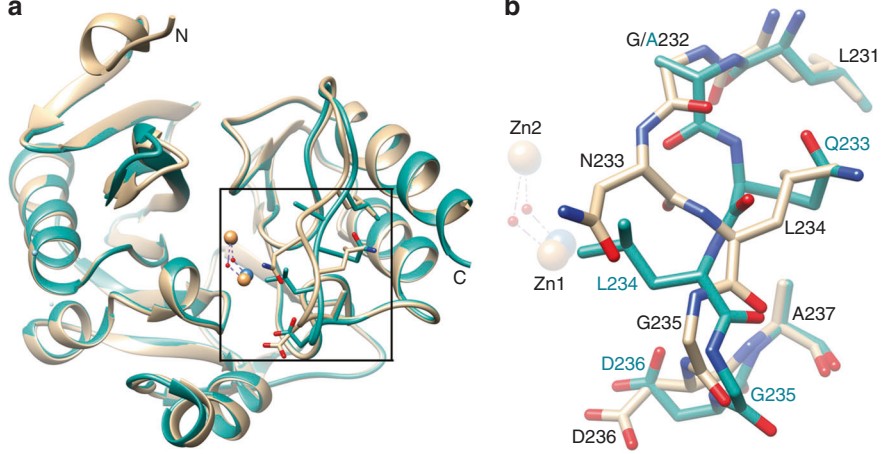

**Fig. 6** Crystal structure of the NDM-1 K224R/G232A/N233Q mutant. **a** Left panel, alignment of structure of wild-type NDM-1 (tan, PDB ID: 3SPU[18]) with that of NDM-1 K224R/G232A/N233Q (dark cyan). The zinc ions are represented as orange and gray spheres in the wild-type and mutant NDM-1 structures, respectively. The water molecule between Zn1 and Zn2 is shown as a red sphere. The side chains of residues 232–237 are shown within the boxed region of each structure. The box indicates the region of the structure shown in detail in panel B. The N- and C-termini are labeled. **b** Boxed region from panel (**a**) showing alignment of the structure of wild-type NDM-1 (tan, PDB ID: 3SPU) with that of NDM-1 K224R/G232A/N233Q (dark cyan) for residues 231 to 237. A large change in the conformation of the 231–237 region of the NDM-1 K224R/G232A/N233Q mutant is seen versus the structure of wild-type NDM-1

with a root-mean-square deviation of 0.304 Å (Supplementary Fig. 5). Substantial variation exists in the L3 loop bordering the active site (Supplementary Fig. 5). This may reflect the flexibility of the loop L3, which has been reported to undergo conformation changes when substrate is bound to the wild-type NDM-1 enzyme[12,23]. The mutations of the triple mutant are in loop L10, which surrounds the active site. The overall conformation of L10 is similar for the four chains except that main chains of Ala232 and Gln233 in chain B vary by about 1 Å from the other chains, which may be due to high flexibility of the region. Flexibility in this region is consistent with high B factor values for Ala232 (36.7 Å$^2$ on average of four chains) and Gln233 (37.5 Å$^2$ average). In addition, the conformation of Arg224 varied among the four chains in that the side chain of the residue in molecules A and D orientates toward the active site and that of B and C is away from the active site (Supplementary Fig. 5).

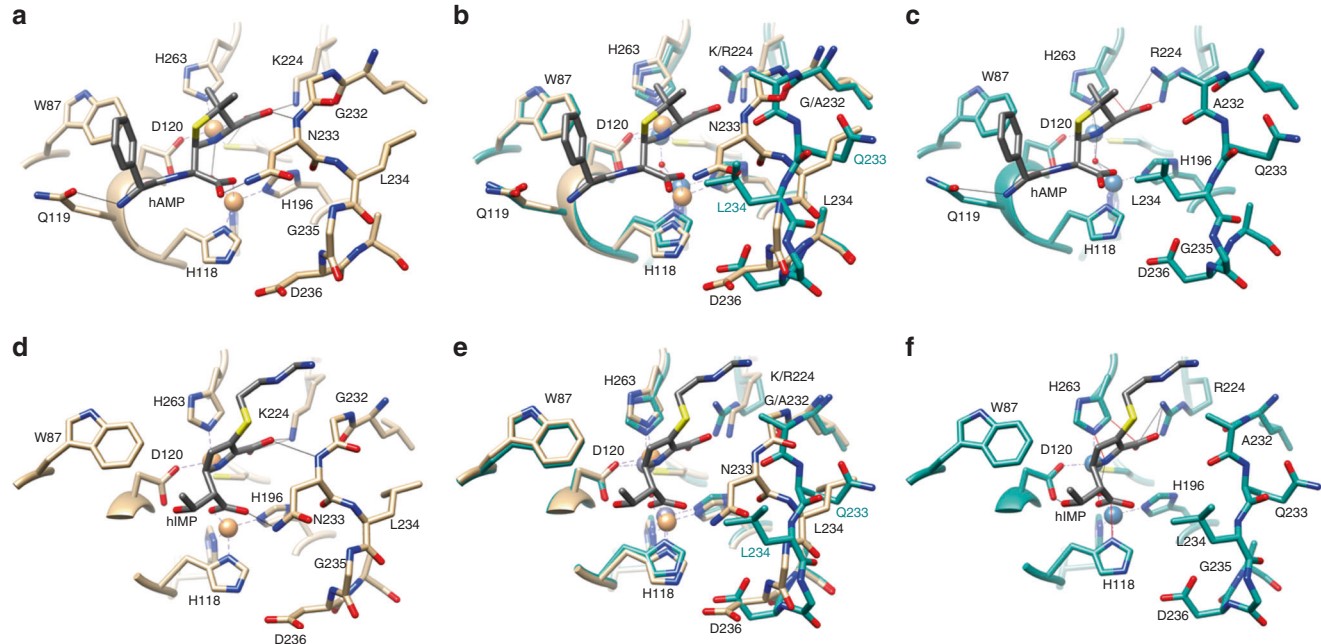

**Fig. 7** Structural alignments of the NDM-1 triple mutant with NDM-1/product complexes. **a** Structure of wild-type NDM-1 (tan) in complex with hydrolyzed ampicillin (3Q6X). Hydrolyzed ampicillin is shown in gray. Hydrogen bonds are indicated by thin black lines. **b** Structural alignment of wild-type NDM-1 (tan) with hydrolyzed ampicillin (gray) and the NDM-1 triple mutant (dark cyan). Hydrogen bonds are not shown for clarity. **c** Structural alignment from panel (**b**) with the wild-type NDM-1 structure removed but leaving the hydrolyzed ampicillin from the wild-type structure in place. **d** Structure of wild-type NDM-1 (tan) in complex with hydrolyzed imipenem (5YPI[7]). Hydrolyzed imipenem is shown in gray. **e** Structural alignment of wild-type NDM-1 (tan) with hydrolyzed imipenem (gray) and the NDM-1 triple mutant (dark cyan). Hydrogen bonds are not shown for clarity. **f** Structural alignment from panel (**e**) with the wild-type NDM-1 structure removed but leaving the hydrolyzed imipenem from the wild-type structure in place. Thin black lines indicate hydrogen bonds. Thin red lines indicate close contacts with steric clash

Chain A of the NDM-1 K224R/G232A/N233Q structure was used for comparison with the apo-NDM-1 wild-type structure (PDB ID: 3SPU)[23]. The K224R/G232A/N233Q mutations do not result in large changes in the overall structure of NDM-1 but do cause a large change in the conformation of loop L10 where the mutations occur (Fig. 6a). Specifically, although the side chain of Arg224 in NDM-1 K224R/G232A/N233Q chain A has a similar orientation as that of Lys224 in the wild-type protein, the side chain of Gln233 in the mutant adopts a very different orientation from that of Asn233 in the wild-type enzyme and is pointed out of the active site (Fig. 6a). The main chain conformation of the triple mutant is drastically altered between Ala232 and Asp236 (Fig. 6b). This has the effect of placing the side chain of Leu234 of the mutant in the position of Asn233 in wild-type and the side chain of Gln233 of the mutant in the position of Leu234 in wild-type. As Asn233 in wild-type NDM-1 is predicted to form a hydrogen bond with the carboxylate group of the reaction intermediate of β-lactams, this interaction would be abolished in the K224R/G232A/N233Q mutant, with the Asn233 hydrogen-bonding group replaced by the Leu234 hydrophobic group. This is further illustrated by superimposing the structure of apo-NDM-1 K224R/G232A/N233Q on the available structures of wild-type NDM-1 complexed with hydrolyzed AMP (PDB:3Q6X)[12] and IMP (PDB:5YPI)[13] where the orientation of the Gln233 side chain is seen oriented away from the hydrolyzed AMP and IMP and is replaced by the hydrophobic Leu234 side chain (Fig. 7). In addition, the methyl group of Ala232 in the mutant protein protrudes toward the active site (Fig. 7b, e), which may cause steric clash with β-lactams with bulky R2 groups such as cephalosporins and carbapenems. Indeed, the G232A single

mutant results in an increase in $K_M$ for IMP from 65 μM to a not measurable value ( > 600 μM) with only a very modest (1.3-fold) increase in $K_M$ for AMP hydrolysis (Table 3). Attempts to obtain the structure of the K224R/G232A/N233Q mutant in complex with AMP, CTX, or IMP products by co-crystallization, as well as soaking were not successful.

## Discussion

As a broad spectrum and readily transferable MBL, NDM-1 confers resistance to nearly all β-lactam antibiotics in a wide range of Gram-negative pathogens[24,25]. NDM-1 has broad specificity with $k_{cat}/K_M$ values of $10^6$–$10^7$ M$^{-1}$ s$^{-1}$ for hydrolysis of penicillins, cephalosporins, and carbapenems[18]. This is in contrast to serine active site β-lactamases such as TEM-1 that efficiently hydrolyzes AMP ($k_{cat}/K_M = 3.0 \times 10^7$ M$^{-1}$ s$^{-1}$) but essentially does not hydrolyze IMP[26]. Deep sequencing analysis of the antibiotic-resistant clones identified both essential and context-dependent residue positions. The former represents a core set of residue positions where the wild-type residue is indispensable for NDM-1 function against any of the tested drugs. The context-dependent residue positions are those where the wild-type residue is required for NDM-1-mediated resistance/hydrolysis of IMP and/or CTX but not AMP. In addition, several non-essential residues were identified where multiple amino-acid substitutions are tolerated without compromising NDM-1 function against any tested β-lactam.

Non-essential residue positions are located largely on the periphery of the active site and not positioned to interact with zinc ions or bound substrate (Fig. 4). Essential core residue

positions were found to be zinc ligand residues or charged residues that link loop structures together in the active site. The essential zinc ligand residues comprised both the histidine site (His116, His118, and His196) and the cysteine site (Asp120, Cys221, and His263) and their functions have been well documented by site-directed mutagenesis studies[22]. In addition, residue positions Asp84, Asp199, and Lys121, whose function in NDM-1 is poorly understood, were also identified to be essential for NDM-1 function for all antibiotics tested (Fig. 3 and Supplementary Data 1). Examination of the NDM-1 structure reveals that both side chain and main chain groups of these residues form hydrogen bonds with multiple residues and serve to link loop structures in the active site[12] (Supplementary Fig. 6). Asp84 forms hydrogen bonds with His55, Val113, Thr115, and Lys121. Asp199 interacts with Ala117, Ser141, and Thr197, whereas Lys121 form hydrogen bonds with His118 and Ser262[12] (Supplementary Fig. 6). The metal-binding sites of MBLs are entirely built on loops and turns. The inter-loop interactions mediated by the Asp84, Asp199, and Lys121 residues may play a critical role in maintaining the position of the loops and stability of NDM-1. The contribution of residues in connecting networks of interactions has been recognized as an important component of protein structure[27,28].

The IMP-1 MBL, similar to NDM-1, is a subclass B1 enzyme with broad specificity and high clinical relevance[3]. Previous saturation mutagenesis studies at 29 residue positions in and near the active of IMP-1 also revealed the zinc-chelating residues are essential for hydrolysis of penicillins, cephalosporins, and carbapenems[29]. The study also identified Asn233 as a substrate-specific residue important for carbapenem but not penicillin hydrolysis. The results for other positions, however, are difficult to compare directly with the NDM-1 results because of the limited number of resistance mutants sequenced in the IMP-1 study[29].

In contrast to subclass B1 enzymes, such as NDM-1 and IMP-1, subclass B2 enzymes such as CphA contain a single zinc (Zn2) and are inhibited on binding of a second zinc[3,16]. In addition, they hydrolyze only carbapenems and are proposed to do so through an altered catalytic mechanism compared with di-zinc enzymes such as NDM-1[30,31]. We previously generated randomized, single codon libraries for 26 residues in and near the active site of CphA and performed deep sequencing of mutants from each library after selection for imipenemase function[32]. A comparison of the results as indicated by sequence logos reveals many of the equivalent positions do not tolerate substitutions and are thus required for IMP hydrolysis for both enzymes (Supplementary Fig. 7). Strikingly, for three positions, the wild-type residue is required for both enzymes but the identity of the residue is different for NDM-1 versus CphA. These positions include Asp/Gly84 (NDM/CphA), His/Asn116, and Gly/Asn220 (Supplementary Fig. 7). Thus, the residue type required for function is dependent on whether the position is in the context of the CphA or the NDM-1 enzyme. Such context-dependent sequence requirements indicate epistasis at these positions. These observations suggest that these positions may control the narrow specificity of CphA. In this regard, it has been reported that a N116H/N220G double mutant changes the substrate specificity of CphA to include penicillins and cephalosporins, similar to subclass B1 enzymes such as NDM-1[33].

Similar to subclass B1 enzymes, such as NDM-1, subclass B3 enzymes contain two zincs in the active site and hydrolyze a broad range of penicillins, cephalosporins, and carbapenems[3,16]. The B3 enzymes, however, contain differences in the active site including an altered Zn2-binding site where the Cys221 ligand in B1 is replaced by Ser221, and instead His121 binds Zn2 in B3 enzymes. Although shifted in position compared with B1

enzymes, mutagenesis studies show His121 is critical for B3 enzyme function, as is Cys221 in NDM-1[34]. In addition, Lys224, which binds the substrate C3/C4 carboxylate in B1 enzymes is absent in B3 enzymes and is replaced by Ser221 and Ser/Thr223[35,36] (Supplementary Fig. 8). Interestingly, saturation mutagenesis studies of the Ser221 and Thr223 residues in the B3 AIM-1 enzyme, which are positioned to bind the C3/C4 carboxylate in a similar role as Lys224 in NDM-1, also are more tolerant to substitutions for AMP hydrolysis compared with IMP hydrolysis, similar to what is observed for Lys224 in this study[37]. Further, B3 enzymes do not contain an equivalent residue as Asn233 in NDM-1 but Gln157, which originates from a different loop, has its terminal amide group in a similar position as that of Asn233 in NDM-1[38] (Supplementary Fig. 8). Mutagenesis of Gln157 in AIM-1 also shows similar results as that for Asn233 in NDM-1, with tolerance to substitutions for AMP hydrolysis but more stringent requirements for IMP[37]. Indeed, saturation mutagenesis of eight active site positions in AIM-1 show the general trend of relaxed sequence requirements for AMP hydrolysis compared with IMP, as observed here for NDM-1[37]. Although this is a limited comparison, it suggests B3 enzymes may also require more sequence information for carbapenem versus penicillin hydrolysis.

The interesting question that arises from the NDM-1 experiments is why does IMP hydrolysis require more precise sequence information in the active site for efficient catalysis than CTX and, in particular, AMP hydrolysis? A characteristic feature of carbapenems, including IMP, is the chirality of the hydroxyethyl group at the C6 position is S, whereas the acyl-amide side chain of penicillins and cephalosporins is R. This leads to different interactions between carbapenems and the active site versus penicillins and cephalosporins. It has been noted that the altered chirality forces the C6 carboxylate formed after bond cleavage of carbapenems to orient differently than that for penicillins and cephalosporins. In available structures with hydrolyzed IMP and meropenem, the carboxylate is shifted to a position between Zn1 and Zn2 where it displaces the bridging water[11,13]. The different binding modes of carbapenems versus penicillins and cephalosporins could result in different sequence requirements for binding and catalysis, as reflected in the deep sequencing results[13]. Also along these lines, based on available structural information, carbapenems make fewer contacts with the enzyme than penicillins and cephalosporins (Supplementary Figs. 9–12)[10–12]. Therefore, the contacts that do occur with carbapenems may be critical and very sensitive to perturbation. The structure of hydrolyzed IMP bound to NDM-1 shows that the R2 group makes minimal contacts with the enzyme, whereas the R1 hydroxyethyl group interacts with the hydrophobic portion of the active site via contact with Trp87[13] (Supplementary Fig. 12). In contrast, AMP makes multiple hydrophobic interactions with Leu59, Met61, and Trp87, as well as a hydrogen bond with Gln119[11,12] (Supplementary Fig. 9). The multiple interactions of the AMP side chain with the enzyme may buffer changes at individual residues making the enzyme more robust to substitutions with respect to AMP hydrolysis.

The more stringent sequence requirements for IMP hydrolysis were further highlighted by the construction of a triple mutant, K224R/G232A/N233Q, based on the sequencing information that exhibited wild-type $k_{cat}/K_M$ for AMP but a 600-fold decrease in IMP hydrolysis.

Lys224 is a conserved residue that facilitates β-lactam substrate binding to the active site through an electrostatic interaction with the negatively charged C3/C4 carboxylate group common to β-lactam antibiotics[3]. The crystal structure of the K224R/G232A/N233Q triple mutant shows the side chain of Arg224 is positioned to interact with substrate but does not provide a clear

rationale for why the substitution negatively affects carbapenem hydrolysis compared with other substrates.

Gly232 and Asn233 are conserved residues that reside on the critical L10 loop flanking the active site of NDM-1[12,23]. Based on structures of NDM-1 in complex with hydrolyzed β-lactams, Gly232 makes hydrophobic interactions with the R2 group of carbapenems and cephalosporins but not with AMP where there is a dimethyl group at C-2 rather than an R group[10–12] (Fig. 2a). The methyl group of Ala232 may clash or limit the flexibility of the R-2 groups of carbapenems and cephalosporins but not the dimethyl group of AMP. Crystal structures also indicate the side chain of Asn233 hydrogen bonds to the newly formed carboxylate group resulting from cleavage of the β-lactam ring, implying that it participates in β-lactam catalysis[10–12]. In the triple mutant, the side chain of Gln233 flips away from the carbonyl oxygen of the substrate and carboxylate of the product (Fig. 6), which would eliminate the hydrogen bond, suggesting this interaction is important for carbapenem but not penicillin catalysis. This is consistent with previous observations that a N233A mutant of NDM-1 retains high-level activity toward AMP but exhibits decreased activity toward carbapenems. Further, changes in the orientation of the main chain for positions 232–236 in the triple mutant places the side chain of Leu234 near the carboxylate group of hydrolyzed AMP and IMP (Fig. 7b, e), which could affect substrate positioning relative to Zn1. As noted above, this carboxylate group is in an altered position for the structures of NDM-1 complexed with hydrolyzed IMP and meropenem compared with structures with hydrolyzed AMP and ceftriaxone due to the altered stereochemistry of the hydroxyethyl group of carbapenems (Fig. 7a, d). Thus, the loss of the Asn233 hydrogen bond could differentially affect carbapenems versus penicillins and cephalosporins.

The NDM-1 enzyme exhibits a broad substrate profile with $k_{cat}/K_M$ values for penicillins, cephalosporins, and carbapenems in the $10^6$–$10^7\,M^{-1}\,s^{-1}$ range[18]. NDM-1 emerged in 2008 from a patient infected with *E. coli* that encoded the enzyme[6]. The ultimate bacterial origins and selective pressures that led to the evolution of the broad substrate profile, however, are not known. One possibility is that the broad substrate profile was shaped by frequent exposure and selective pressure from all three classes of β-lactam antibiotics. The results presented here, however, suggest the possibility that the broad substrate profile could have evolved from selective pressure exclusively from carbapenems in that the residue positions and sequences required for penicillin and cephalosporin hydrolysis are a complete subset of those required for carbapenem hydrolysis, i.e., an NDM active site poised for carbapenem hydrolysis is also competent for penicillin and cephalosporin hydrolysis.

In summary, deep sequencing of random mutant libraries of NDM-1 β-lactamase selected for β-lactam antibiotic resistance together with biochemical analyses has shown that there is a core set of residues that contribute strongly to the hydrolysis of all β-lactam substrates but that additional residues are required to attain levels of IMP hydrolysis similar to that observed for CTX and AMP. Thus, hydrolysis of IMP is a more difficult task than hydrolysis of AMP or CTX. A practical implication of the findings of this study is that, if an inhibitor of NDM-1 is developed, it should be used in combination with a carbapenem because the stringent sequence constraints for carbapenem hydrolysis will restrict the number of substitutions that are possible and are able to confer inhibitor resistance and retain efficient carbapenem hydrolysis.

## Methods

**Bacterial strains and plasmids.** *E. coli* XL1-Blue (Stratagene) and *E. coli* BL21 (DE3)[39] were used as the host strains for the construction of codon randomization

libraries and for over-production of wild-type and mutant NDM-1 enzymes, respectively. The plasmid pTP470, which encodes chloramphenicol resistance and is a *lacI$^q$*-deletion derivative of pTP123[40], contains the gene for NDM-1, whose expression is under the control of the isopropyl β-D-1-thiogalactopyranoside (IPTG)-inducible *trc* promoter. The Strep-tagII sequence is included at the C-terminus of NDM-1 to allow for monitoring the expression of NDM-1 in *E. coli* by immunoblotting with anti-Strep-tagII antibody[41,42]. For expression of mature NDM-1, a truncated *NDM-1* gene encoding mature NDM-1 enzyme (mNDM-1, Gly$^{36}$ to Arg$^{270}$)[18] was cloned between *Nde*I and *Xho*I restriction sites of a modified pET28a vector (Novagen), in which the thrombin recognition sequence was replaced with the tobacco etch virus (TEV) protease recognition sequence. Site-directed mutagenesis was performed on NDM-1-StrepII-pTP470 and mNDM-1-pET28a-TEV to obtain expression vectors for NDM-1-StrepII and His-NDM-1 mutants[43].

**Construction of NDM-1 single codon randomization libraries.** NDM-1 single codon random libraries were constructed by oligonucleotide directed mutagenesis[32]. First, to eliminate any wild-type NDM-1 background, a *Xho*I restriction site was inserted near the target codon for randomization by oligonucleotide directed mutagenesis (Supplementary Table 2). The insertion of the *Xho*I recognition sequence was designed to also introduce a frameshift mutation in the NDM-1 gene to ensure the insert mutant is non-functional. The *Xho*I insert mutant was then used as the template for randomization of each codon by using partially overlapping (25 base pair (bp)) primers for PCR amplification (Supplementary Table 2). The codon for the target residue was substituted by NNS (where N is any of the four nucleotides and S is G or C) so that codons for all 20 amino acids were represented in the library. The resulting mutagenesis reactions were treated with the *Dpn*I and *Xho*I restriction enzymes to eliminate non-mutagenized plasmids and used for transformation into *E. coli* XL1-Blue by electroporation. A minimum of 300 colonies were pooled for each library construction and used for preparation of plasmid DNA to obtain a single codon randomization library.

**Selection of library mutants with resistance to β-lactams.** In order to select for functional NDM-1 mutants from each library, 100 ng of DNA from each plasmid library was transformed into *E. coli* XL1-Blue by electroporation and the bacterial cells were spread on Luria-Bertani (LB) agar plates containing 0.5 mM IPTG, 12.5 μg ml$^{-1}$ chloramphenicol, and either 100 μg ml$^{-1}$ AMP, 20 μg ml$^{-1}$ CTX, or 1 μg ml$^{-1}$ IMP. These levels of antibiotics were determined to select for clones with near wild-type β-lactamase activity. The transformed bacteria were also spread on the LB agar plates containing 0.5 mM IPTG and 12.5 μg ml$^{-1}$ chloramphenicol as naive library control experiments. The transformation cultures were diluted appropriately to produce approximately 1000 colonies on each selection plate. The resulting colonies were pooled and used for plasmid preparation.

**Preparation of NDM-1 library samples for deep sequencing.** The plasmid DNA obtained from pooled colonies for each library after each antibiotic selection was used as template DNA for PCR reactions in preparation for deep sequencing. PCR primers for amplification were designed to amplify the region of *bla$_{NDM-1}$* containing the randomized codon of interest (Supplementary Table 2). The PCR primers also contained a 7-bp barcode sequence that was unique for each pooled library from each drug selection. The barcode sequences differ from each other by at least 2 bp and the primers were designed so that the PCR amplicons were approximately 150 bp in length. PCR products were purified from a 1.5% agarose gel using QIAquick gel extraction kit (Qiagen) and the DNA concentration of each sample was quantified using a Nanodrop instrument. The resulting purified PCR products from each library and each antibiotic selection, as well as each naive library were then pooled into a single tube. The pooled PCR products were ligated with adapters for sequencing and Illumina paired-end MiSeq sequencing (2 × 150-bp read length) was performed by the Human Genome Sequencing Center at Baylor College of Medicine.

**Analysis of Illumina deep sequencing data.** Illumina MiSeq sequencing returned FASTQ files containing the sequencing reads and quality information. The sequencing data were quality checked using the Galaxy web server (https://usegalaxy.org/), which showed that sequence quality was high (scores over 30) for all nucleotide positions except those at the extreme end of the reads (Supplementary Fig. 13). A custom Perl script was used to extract the mutant sequences for each naive library and each antibiotic selection by using matches to the appropriate barcode sequence, as well as the sequences 10-bp upstream and downstream of the randomized codon[20,32]. This yielded $3.1 \times 10^7$ reads for the 124 experiments (31 libraries × 4) with $2.5 \times 10^5$ reads for each experiment on average. Because the randomization codons for all libraries were located 30–40 bp from the 5′ terminus of each PCR product and thus in a region of high quality, filtering was not performed on the sequencing data. A custom Python script was then used to translate codons to amino-acid sequences and count the occurrence of each amino-acid type in each experiment, as shown in Fig. 2b and Supplementary Data 1.

**Creation of sequence logos.** In order to represent the predominant amino-acid sequence types in the libraries after selection for antibiotic resistance, sequence

logos were created[32,44]. The logos are based on the frequency of occurrence of each amino acid in each experiment. However, for the Gly65, Gly195, and Gly232 libraries in which glycine was represented at a significantly higher frequency in the naive libraries than other amino-acid types, the frequency of occurrence of each amino acid in the antibiotic selection experiments was normalized to their frequency of occurrence in the naive libraries to eliminate the library coverage bias or sequencing bias.

**Determination of β-lactam antibiotic resistance levels.** NDM-1 and its mutants were expressed in *E. coli* XL-1 Blue. The antibiotic resistance levels for AMP, CTX, and IMP were determined by measuring the MIC of each drug for *E. coli* containing NDM-1 wild-type or mutants using serial twofold dilutions of antibiotics. For this purpose, 1:100,000 diluted overnight bacterial cultures were mixed with serial twofold dilutions of antibiotics in 96-well plates in a final volume of 0.2 ml. After 16-h incubation at 37 °C with shaking, bacterial growth was recorded by reading the absorbance of the cultures at 600 nM (OD$_{600}$) with a Tecan microplate reader. The MIC was defined as the lowest concentration of antibiotic that inhibits 90% of cell growth (IC$_{90}$), compared with a control culture without the antibiotic. The plasmid encoding NDM-1 contains an IPTG-inducible *trc* promoter and 0.2 mM IPTG was included in the culture medium for IMP MIC determinations. However, for AMP and CTX MIC determinations, IPTG was not included as the level of uninduced expression of NDM-1 was sufficient for high-level resistance against these antibiotics (4096 and 256 μg ml$^{-1}$, respectively).

**Relative fitness calculations.** For each library, relative fitness was used to examine the relationship between antibiotic resistance levels of NDM-1 mutants compared with their frequency of occurrence in the antibiotic selection experiments. For deep sequencing results of the libraries with or without antibiotic selections, the relative fitness $F_u^a$ of amino-acid substitution $a$ at each position $u$ was calculated as described by Stiffler et al.[40] using the equation below:

$$F_u^a = \log_{10}\left[\frac{N_u^{a,sel}}{N_u^{a,naive}}\right] - \log_{10}\left[\frac{N_u^{wt,sel}}{N_u^{wt,naive}}\right] \quad (1)$$

where $N_u^{a,sel}$ and $N_u^{a,naive}$ represent the mutant allele count of amino acid $a$ at position $u$ in the selected and unselected (naive) residue position library and $N_u^{wt,sel}$ and $N_u^{wt,naive}$ represent the wild-type allele count in the selected and naive residue at position $u$ in the library, respectively. Basically, the $F_u^a$ value reflects the enrichment or depletion of mutant alleles under selection relative to that of the wild-type allele. An $F_u^a$ value close to 0 indicates the mutation occurs at equal frequency to the wild-type amino acid at this position. A negative $F_u^a$ value shows that the substitution is less fit and occurs less frequently than the wild-type residue at this position. A positive value, however, indicates that this amino acid occurs more often upon selection than the wild-type residue such that the substitution increases the fitness.

For the results of antibiotic resistance determinations, the relative fitness $f_u^a$ of amino-acid substitution $a$ at each position $u$ was calculated using the equation below:

$$f_u^a = \log_{10}\left(MIC_u^a\right) - \log_{10}\left(MIC_u^{wt}\right) \quad (2)$$

where $MIC_u^a$ and $MIC_u^{wt}$ represent the MIC values of the mutant allele and the wild-type allele, respectively. An $f_u^a$ value of 0 indicates the amino-acid substitution does not change antibiotic resistance function of NDM-1 β-lactamase. A negative $f_u^a$ value indicates that the amino-acid substitution decreases the antibiotic resistance function of the β-lactamase, whereas a positive $f_u^a$ value indicates that the substitution increases the antibiotic resistance function of the β-lactamase.

**Calculation of effective number of substitutions ($k^*$).** The effective number of substitutions ($k^*$) at each position was calculated using the equations below, where $S$ is the entropy, $p_i$ is the fraction of times the $i$th type appears at a position, and $k$ is the number of different amino-acid residue types that appear at a position[20,21]. A $k^*$ value of one indicates that only one amino acid is observed at this position while a value of 20 indicates that all amino acids are tolerated equally.

$$S = -\sum_{i=1}^{k} p_i\left(\log_2 p_i\right) \quad (3)$$

$$k^* = 2^S \quad (4)$$

**Protein expression and purification.** The mNDM-1-pET28a-TEV plasmid encoding His-tagged-NDM-1 enzymes was used for protein expression in *E. coli* BL21 (DE3) cells and subsequent purification for enzyme kinetics and protein crystallization experiments. The *E. coli* cells containing the NDM-1 expression plasmid were grown in LB medium containing 25 μg ml$^{-1}$ kanamycin. Expression of the His-tagged-NDM-1 protein was induced in mid-log-phase cultures with 0.5 mM IPTG at 20 °C for 16 h. The cells were pelleted and suspended in lysis buffer (20 mM HEPES, 500 mM NaCl, 20 mM imidazole, pH 7.4) and lysed using a

French press. After centrifugation, soluble fractions in the supernatant were loaded onto a HisTrap$^{TM}$ FF column (GE Healthcare) and His-tagged-NDM-1 was eluted with an imidazole gradient in the lysis buffer. The His-tag was removed by incubating with TEV protease for 24 h at 4 °C at a ratio of 1:50. The TEV protease was removed by passing the preparation through the HisTrap$^{TM}$ FF column again. NDM-1 β-lactamase in the column flow through fraction was concentrated with Amicon concentrator units (EMD Millipore) and further purified by gel-filtration chromatography using a Superdex 75 GL 16/600 sizing column (GE Healthcare) with 20 mM HEPES, 100 mM NaCl, 50 μM ZnSO$_4$, pH 7.4 as running buffer. Fractions containing NDM-1 were pooled and concentrated. NDM-1 mutants were purified and quantified in the same way as that for wild-type NDM-1. The purity of wild-type NDM-1 and each of the mutant enzymes was > 90% based on sodium dodecyl sulfate polyacrylamide gel electrophoresis (SDS-PAGE) analysis. The final protein concentration of wild-type NDM-1 and mutants was determined by measuring absorbance at 280 nm with a DU800 spectrophotometer (Beckman Coulter) and using an extinction coefficient of $\varepsilon_{280} = 27,690$ M$^{-1}$ cm$^{-1}$, which was calculated using the ExPASy ProtParam tool.

**Determination of kinetic parameters.** Enzyme kinetic parameters for the hydrolysis of AMP, CTX, or IMP by wild-type or mutant NDM-1 β-lactamase was performed at 25 °C in 50 mM HEPES (pH 7.4) supplemented with 10 μM ZnSO$_4$, in which NDM-1 exhibits maximal activity with β-lactam antibiotics[18]. Antibiotic hydrolysis was monitored with a DU800 spectrophotometer (Beckman Coulter) equipped with a thermostatically controlled cell by following the absorbance change of AMP at 235 nm ($\Delta\varepsilon_{235nm} = -900$ M$^{-1}$ cm$^{-1}$), CTX at 260 nm ($\Delta\varepsilon_{260nm} = -7250$ M$^{-1}$ cm$^{-1}$), and IMP at 300 nm ($\Delta\varepsilon_{300nm} = -9000$ M$^{-1}$ cm$^{-1}$). Cuvettes with 0.1- or 1-cm path lengths were used, depending on the substrate concentration being examined. For wild-type and most NDM-1 mutants, $k_{cat}$ and $K_M$ parameters were determined under initial-rate conditions by fitting the initial velocity ($v_o$) at various substrate concentrations to the Michaelis–Menten equation ($v = V_{max} [S]/(K_M + [S])$) using GraphPad Prism5. When $V_{max}$ could not be determined because $K_M$ was too high, the catalytic efficiency ($k_{cat}/K_M$) was determined by analyzing the complete hydrolysis time courses at low antibiotic concentration and fitting the data to the equation $v = k_{cat}/K_M$ [E][S][32,45,46]. Kinetic parameters were averaged from at least two independent determinations. The background rate of hydrolysis of each of the substrates in the absence of enzyme was undetectable over the time frame of the initial velocity determinations.

**Protein expression level determination.** The effect of amino-acid substitutions on steady-state expression levels of NDM-1 β-lactamase in *E. coli* was determined by Western blotting[32,42]. For this purpose, overnight cultures of *E. coli* XL1-Blue strains containing wild-type or mutant NDM-1-StrepII were diluted 1:100 into LB medium containing 12.5 μg ml$^{-1}$ chloramphenicol and grown at 37 °C with shaking until the OD$_{600}$ reached 0.6–0.9. Cells were collected by centrifugation and lysed in B-PER (Thermo Scientific) containing 0.1 mg ml$^{-1}$ lysozyme and 0.02 mg ml$^{-1}$ DNaseI, whose volume was adjusted according to the OD$_{600}$ of the culture to ensure that same cell density was obtained. The same volumes of cell lysates were subjected to SDS-PAGE and used for Western blotting. Because a StrepII tag was fused to the C-terminus of wild-type or mutant NMD-1, their expression was detected by probing with a horseradish peroxidase (HRP)-conjugated mouse monoclonal anti-StrepII antibody (Novagen). In addition, the same membrane was also probed with an antibody against DnaK (Enzo Life Sciences), which serves as a loading control. The hybridization signal was quantified by densitometry using ImageJ software (NIH).

**X-ray structure determination of NDM-1 triple mutant.** For crystallization of NDM-1 K224R/G232A/N233Q, the protein was purified as a His-tagged version by using HisTrap$^{TM}$ FF column (GE Healthcare). After removal of the His-tag by TEV protease, NDM-1 K224R/G232A/N233Q was further purified using gel-filtration chromatography using a Superdex 75 GL 16/600 sizing column (GE Healthcare) with 20 mM HEPES, 150 mM NaCl, 2 mM DTT as running buffer. The protein was concentrated to 40 mg ml$^{-1}$ and was screened for crystallization using commercially available crystal screens from Hampton Research and Qiagen. A Mosquito automated nanoliter liquid handler robot (TTP LabTech) was utilized to mix protein solution (0.1 μl) and the reservoir solution (0.1 μl). The mixture was left to equilibrate against the reservoir solution (70 μl) at 25 °C by using the hanging-drop method. Crystals formed within a few days under various crystallization conditions. For data collection, crystals were soaked in cryoprotectant solution (25% glycerol diluted in reservoir solution) for 30 s and flash frozen in liquid nitrogen. A 1.65 Å data set was collected on the 8.2.1 beamline (1 Å wavelength) at the Advanced Light Source synchrotron in Berkeley, CA for the crystal grown with 0.2 M LiCl, 0.1 M HEPES, pH 7.0, 20% (w/v) PEG6000.

Diffraction data for crystals of the apo-β-lactamase and its CTX complex of NDM-1 K224R/G232A/N233Q were indexed, integrated and scaled using HKL2000[47] (Table 4). The structures were solved by molecular replacement using the program Phaser[48] with chain A of AMP bound NDM-1 structure as a starting model (PDB ID: 3Q6X)[12]. The structures of apo protein and protein/CTX were solved to 1.75 Å and 1.60 Å, respectively. However, no density of CTX was observed in the active site of the molecule for the protein/CTX structure. The apo

| Table. 4 Data collection and refinement statistics for NDM-1 K224R/G232A/N233Q | |
|---|---|
| Data collection | |
| Space group | P1 |
| Cell dimensions | |
| $a$, $b$, $c$ (Å) | 46.17, 68.86, 68.44 |
| $\alpha$, $\beta$, $\gamma$ (°) | 92.23, 77.03, 91.84 |
| Resolution (Å) | 38.06–1.75 (1.81–1.75)[a] |
| $R_{merge}$ | 0.053 (0.325)[a] |
| $I/\sigma I$ | 17.6 (3.3)[a] |
| Completeness (%) | 91.07 (54.76)[a] |
| Redundancy | 2.3 (2.2)[a] |
| Refinement | |
| Resolution (Å) | 38.06–1.75 |
| No. reflections | 75370 |
| $R_{work}/R_{free}$ | 0.1554/0.1990 |
| No. atoms | |
| Protein | 6807 |
| Zn | 8 |
| Water | 684 |
| $B$-factors | |
| Protein | 18.68 |
| Ligand/ion | 27.06 |
| Water | 28.75 |
| R.m.s. deviations | |
| Bond lengths (Å) | 0.006 |
| Bond angles (°) | 0.78 |

Four molecules in the asymmetric unit
[a]Values in parentheses are for highest-resolution shell

protein structure was further refined by using PHENIX[49] with several rounds of manual remodeling in COOT[50] between refinement cycles (Table 4). Simulated annealing was performed on the partially refined model using PHENIX[49]. Ramachandran statistics for the final refined structure are 98.24% favored and 0.22% outliers. The fully refined structure has been deposited in the Protein Data Bank with the entry code 6C89. Structure figures were generated using Chimera[51].

**Code availability**. The custom scripts used to process the next-generation sequencing data as described above are available upon request from the corresponding author.

## Data availability

Coordinates and structure factors have been deposited in the Protein Data Bank under accession code 6C89. All relevant data associated with the paper are available upon request from the corresponding author.

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

## Acknowledgements

We thank Amy Hurwitz for assistance in analysis of DNA sequencing data, Cameron Brown for assistance in protein crystallization experiments, and Hiram Gilbert for discussions and comments on the manuscript. This research was supported by National Institutes of Health grants R01 AI106863 and AI32956 to T.P. B.V.V.P. acknowledges support from the Robert Welch Foundation (Q1279). The Berkeley Center for Structural Biology is supported in part by the National Institutes of Health, National Institute of General Medical Sciences, and the Howard Hughes Medical Institute. The Advanced Light Source is supported by the Director, Office of Science, Office of Basic Energy Sciences of the U.S. Department of Energy under contract no. DE-AC02-05CH11231.

## Author contributions

Z.S. conceived and performed experiments and wrote the paper. L.H. performed experiments and data analysis. B.S. performed experiments and data analysis. B.V.V.P. conceived experiments and wrote the paper. T.P. conceived experiments, performed data analysis, and wrote the paper.

## Additional information

**Competing interests:** The authors declare no competing interests.

