## [Peer Review File · Nature Communications]

Reviewer #1

Remarks to the Author:

Sun and co-workers used a random mutagenesis approach to evaluate the importance of a panel of active site residues of the metallo-beta-lactamase NDM-1 in the degradation of penicillin, cephalosporin, and carbapenem antibiotics. Plasmid libraries encoding NDM-1 variants with sufficient activity to provide resistance to these antibiotics were screened by deep sequencing, and the extent of amino acid variability observed at each position was taken as an indication of the importance of this residue in enzyme activity (against a particular substrate) or stability. Through these analyses, the authors observed that carbapenem hydrolysis is apparently dependent on a greater number of residues than was the hydrolysis of penicillins or cephalosporins. The impact of representative substitutions on MIC values, kinetic parameters, and enzyme stability was characterised. Three substitutions were combined to obtain an enzyme with wild-type activity against ampicillin but with greatly impaired imipenem (a carbapenem) hydrolytic activity. A crystal structure was solved for this variant, which revealed small changes in the active site.

While the authors have demonstrated that this approach is an interesting way to investigate the importance of particular amino acids in an enzyme active site, and have built up a strong body of data supporting this for NDM-1, I have major concerns about the novelty of this work. The authors employed an apparently identical strategy in 2016 (Sun et al., *Sci. Rep.*, 2016, DOI:10.1038/srep33195) during their investigation of the active site of the subclass B2 metallo-beta-lactamase CphA (although this work was focused on carbapenem hydrolysis, and no crystallographic work was described). Furthermore, 14 years ago, this group has also reported the use of a codon randomization strategy to investigate the importance of active site residues of the subclass B1 metallo-beta-lactamase IMP-1 (i.e., same MBL subclass as NDM-1) for the differential hydrolysis of penicillins, cephalosporins, and carbapenems (Materon et al., *J. Mol. Biol.*, 2004, 344, 653-663). The lack of comparison of their results obtained with NDM-1 to the previously described results for CphA and IMP-1 is a strange omission (notably, the work describing IMP-1 was not referenced, despite at least five of the amino acids proposed as important for IMP-1 activity are noted as being important for carbapenem hydrolysis over hydrolysis of penicillins/cephalosporins by NDM-1).

As the methodology has been previously established, the question is whether the particular results obtained for NDM-1 are sufficiently novel for publication in *Nature Communications*. It is interesting that carbapenem hydrolysis apparently requires more active site residues than other classes of beta-lactams, and the authors have proposed rationales for the roles of these particular residues (in part on the basis of crystallographic analysis). However, this paper would benefit greatly from a deeper investigation of the roles of these residues in governing the interaction of carbapenems with NDM-1, work which I feel would be essential for publication in a high quality journal. Ideally, crystal structures with bound hydrolyzed penicillin/carbapenem (which the authors did attempt to obtain) would more reliably show the interactions of these particular residues with the different substrate types (granted, there are questions whether this would be significant for the unhydrolysed substrate). Overall, I expect that this work would be very well received in a more specialized venue, but do not consider the results presented to merit publication in *Nature Communications*.

Some minor points:

- While the introduction and results sections are clear and well-written, the discussion section would benefit from some revision. As indicated above, the discussion also has some notable omissions.
- Figure 1. Panel B. The mechanism appears to show an asparagine with a charged ammonium group – is this supposed to be the (uncharged) side chain amide? The structure in panel C would benefit from enlargement (or removal, given the overlap with Figure 4).
- Figure 2 – Please add the stereochemistry for the beta-lactams. Please also correct the side-chain of imipenem, which should not have a carbon-carbon double bond

- Figure 6 – His118 is listed twice in panel A
- Table S1 appears to be missing from the Supporting Information
- Page 3 paragraph 4 – make clear the role of polar residues in binding the substrate carboxylate
- Please minimize the 'deep sequencing' jargon
- It would be of interest to define the methods looking at beta-lactams without side chains e.g.
 - clavamate would be of interest
- Are the pH/buffer/temperature dependencies of the mutant enzymes the same as WT? (This information is essential at least for key mutants).
- Label N- and C-termini on crystal figures (4,6)
- Show stereoview electron density maps for active site residues.

Reviewer #2:

Remarks to the Author:

The study represents an interesting analysis of the sequence requirements of NDM-1 to maintain activity for a broad range of β -lactam substrates. The take home message is that in their quest to design NDM-1 inhibitors researchers need to use carbapenem substrates for testing the effects of such compounds. In this context it is somewhat of a pity that the authors did not briefly elaborate on current advances in inhibitor design against MBLs in general, and NDM-1 in particular (see, for instance, recent studies from the Schofield and McGeary groups).

Also, since MBLs from the B1 and B2 subgroups are related in an evolutionary sense, but the B2 representatives have a distinct preference for carbapenem substrates I wonder if any of the residues identified as important for the carbapenemase activity in NDM-1 are conserved in B2 MBLs? A more detailed comparison between NDM-1 and B2 MBLs may enhance the significance and broader impact of the study.

Since MBLs from the B3 subgroup are evolutionarily distinct, but functionally quite similar to B1 MBLs an expansion of the discussion to include a sequence comparison between NDM-1 and at least a couple of well-known B3 MBLs (e.g. L1 and AIM-1) would possibly enhance the impact of the present study even further.

The analysis of deep sequencing data is not an area I have great experience in, but as far as I can evaluate the study has been conducted in proficient manner. The presentation is equally proficient but the text may need some overhaul. Various grammatical errors pervade the manuscript, and the second paragraph in the Discussion section (page 12) appears rather speculative. On occasion (e.g. line 3 on page 12) residues are labeled in single character format (L121), while most of the time the three-letter abbreviation was used. Consistent use of abbreviations and notations should be ascertained throughout the manuscript.

Lastly, is the term "hub residue" commonly used? This question may be a reflection of my ignorance but a brief definition when used first might be useful.

Reviewer #3:

Remarks to the Author:

This manuscript reports a well conducted study related to antibiotic resistance. The hypothesis is sound and the question of interest, although the interest is quite specific to an expert audience. The work is of high quality, the methodology and analysis appear robust. The results of the deep-sequencing analysis comparing the effects of mutations on bacterial survival in the presence of 3 antibiotics is striking and is set in an appropriate light. The correlation between the results of deep sequencing and traditional sequencing appears sufficiently robust to support the conclusions and allows the reader to appreciate the advantage gained by applying that very high throughput strategy. A reasonable number and distribution of mutants were selected for more detailed analysis of the enzyme activity, and those results clearly support the plate screening results.

I disagree in part with the conclusions which I judge to be too general. The study was undertaken with 1 β -lactam antibiotic of each of 3 classes. The results therefore compare these 3 antibiotics. The conclusions state the demonstration of differences for the 3 classes. This was not shown. It can, however, be said that the results for the individual β -lactam antibiotics of each of 3 classes suggests that the observations may extend to the classes.

Part of the discussion illustrates this: 'based on available structural information, carbapenems make fewer contacts with the enzyme than penicillins and cephalosporins (Fig. S4-S6)'; a single β -lactamase-bound inhibitor serves to illustrate the point. It should be toned down to reflect the difference between what is demonstrated or known for a single substrate, and for a class.

The authors addressed this in part (page 10): 'To investigate whether this finding can be generalized to β -lactams in the same class, the triple mutant K224R/G232A/N233Q was tested for hydrolysis of another penicillin and carbapenem, i.e., benzylpenicillin and meropenem, respectively. As shown in Table S2, compared to wild-type NDM-1 enzyme, the triple mutant displays a 25-fold lower k_{cat}/K_M value for hydrolyzing benzylpenicillin but a 250-fold lower k_{cat}/K_M value for hydrolyzing meropenem. Therefore, the result is consistent with the idea that more extensive amino acid sequence information is required in the active site of NDM-1 for carbapenem hydrolysis compared to other β -lactam antibiotics.' The tone here is acceptable: these results are consistent with the hypothesis of generality. Nonetheless, these are results only of one triple mutant and do not allow to conclude on the array of point mutants.

Page 9: amino acid substitutions at nonessential residue positions Phe64 and Asp225: to confirm that these positions can be substituted, it should also be demonstrated that their level of expression was similar to WT.

Methods:

Calculation of effective number of substitutions (k^*) is not clear because π is not defined.

What was the purity achieved for the different mutants that were purified and how was purity accounted for in determination of kinetic parameters (assuming it is not always very high, in the case of poorly expressed mutants for example) ? What is the level of background activity for hydrolysis of each of the substrates in absence of any NDM-1 mutant?

Response to reviewer comments

Responses to the reviewer comments are listed below. Note that the changes described in the responses are indicated by line numbers in the revised manuscript. These changes are also highlighted in yellow in the revised manuscript.

Reviewer 1

Major points:

1. Sun and co-workers used a random mutagenesis approach to evaluate the importance of a panel of active site residues of the metallo-beta-lactamase NDM-1 in the degradation of penicillin, cephalosporin, and carbapenem antibiotics. Plasmid libraries encoding NDM-1 variants with sufficient activity to provide resistance to these antibiotics were screened by deep sequencing, and the extent of amino acid variability observed at each position was taken as an indication of the importance of this residue in enzyme activity (against a particular substrate) or stability. Through these analyses, the authors observed that carbapenem hydrolysis is apparently dependent on a greater number of residues than was the hydrolysis of penicillins or cephalosporins. The impact of representative substitutions on MIC values, kinetic parameters, and enzyme stability was characterised. Three substitutions were combined to obtain an enzyme with wild-type activity against ampicillin but with greatly impaired imipenem (a carbapenem) hydrolytic activity. A crystal structure was solved for this variant, which revealed small changes in the active site.

While the authors have demonstrated that this approach is an interesting way to investigate the importance of particular amino acids in an enzyme active site, and have built up a strong body of data supporting this for NDM-1, I have major concerns about the novelty of this work. The authors employed an apparently identical strategy in 2016 (Sun et al., Sci. Rep., 2016, DOI:10.1038/srep33195) during their investigation of the active site of the subclass B2 metallo-beta-lactamase CphA (although this work was focused on carbapenem hydrolysis, and no crystallographic work was described). Furthermore, 14 years ago, this group has also reported the use of a codon randomization strategy to investigate the importance of active site residues of the subclass B1 metallo-beta-lactamase IMP-1 (i.e., same MBL subclass as NDM-1) for the differential hydrolysis of penicillins, cephalosporins, and carbapenems (Materon et al., J. Mol Biol., 2004, 344, 653-663). The lack of comparison of their results obtained with NDM-1 to the previously described results for CphA and IMP-1 is a strange omission (notably, the work describing IMP-1 was not referenced, despite at least five of the amino acids proposed as important for IMP-1 activity are noted as being important for carbapenem hydrolysis over hydrolysis of penicillins/cephalosporins by NDM-1).

As the methodology has been previously established, the question is whether the particular results obtained for NDM-1 are sufficiently novel for publication in Nature Communications. It is interesting that carbapenem hydrolysis apparently requires more active site residues than other classes of beta-lactams, and the authors have proposed rationales for the roles of these particular residues (in part on the basis of crystallographic analysis). However, this paper would

benefit greatly from a deeper investigation of the roles of these residues in governing the interaction of carbapenems with NDM-1, work which I feel would be essential for publication in a high quality journal. Ideally, crystal structures with bound hydrolyzed penicillin/carbapenem (which the authors did attempt to obtain) would more reliably show the interactions of these particular residues with the different substrate types (granted, there are questions whether this would be significant for the unhydrolysed substrate). Overall, I expect that this work would be very well received in a more specialized venue, but do not consider the results presented to merit publication in Nature Communications.

Response: The reviewer points out that the methodology used was previously established using IMP-1 and CphA β -lactamases, which is correct. With regard to IMP-1, the codon randomization and selection approach was utilized to study sequence requirements for penicillin, cephalosporin and carbapenem hydrolysis. A comparison of that data with the NDM-1 results is now provided on page 14, lines 369-375 of the revised paper. Note, however, that only ~10 sequences were obtained for each selection for the previous IMP-1 study due to the limitations of sequencing technology.

Our NDM data is vastly expanded with an average of 250,000 sequences obtained for each selection experiment. Thus, our NDM results provide a much more comprehensive and quantitative evaluation of the relative impact of each possible substitution at each of the 31 first and second shell residues examined. Thus, our work provides the first comprehensive evaluation of the active site sequence requirements for the important NDM enzyme. We believe this is a significant contribution to understanding NDM-1 and metallo- β -lactamase structure and function.

Another important contribution of our work that extends beyond previous studies is demonstrating that it is possible to “disconnect” carbapenem hydrolysis from penicillin and cephalosporin hydrolysis. This is shown with the K224R/G232A/N233Q triple mutant that was constructed based on the deep sequencing results and hydrolyzes ampicillin at wild-type levels but hydrolyzes imipenem at 600-fold lower levels than wild type. This was not obvious coming into the study as the wild-type enzyme catalyzes the hydrolysis of these classes of antibiotics at similar rates and the results strongly support the idea that, beyond the core residues required for all substrates, carbapenem hydrolysis has unique sequence requirements.

With regard to the previous CphA study, the reviewer makes an excellent point that these should be compared with the NDM-1 results as they both utilized deep sequencing, making the data comparable. **In response to this comment**, a new paragraph has been added to the discussion section (paragraph 5 of discussion, lines 376-392) comparing the results of the NDM study with that of B2 CphA enzyme with respect to carbapenem hydrolysis and the narrow specificity of CphA compared to B1 enzymes like NDM. New Figure S7 shows the comparison using sequence logos. The comparison shows that many of the same active site residues are identified as important for both NDM-1 and CphA carbapenem hydrolysis. Also, strikingly, for three positions (Asp/Gly84 (NDM/CphA), His/Asn116, and Gly/Asn220), the wild-type residue is required for both enzymes but the identity of the residue is different for NDM-1 versus CphA.

This observation indicates epistasis and suggests that these three positions may be important for determining the narrow substrate specificity of CphA compared to NDM-1. Support for this hypothesis comes from previous published studies showing that mutation of two of these positions in CphA broadens the specificity of the enzyme to include penicillins and cephalosporins. In order to facilitate the comparison of results, an additional random library at NDM-1 position Gly220 was constructed for comparison to the CphA Asn220 results. This library was selected on antibiotics, and deep-sequenced. In the revised version, description and discussion of the Gly220 data has been added to the text at lines 145, 170 and 386 and figure 3-5 have been modified to include the new data for Gly220. In addition, the DNA sequencing statistics in lines 116-119 and 528-529 are modified to reflect the addition of the Gly220 library sequencing.

The reviewer further indicates that the paper would benefit greatly from a deeper investigation of the role of the residues in carbapenem hydrolysis, ideally with crystal structures with hydrolyzed penicillin/carbapenem. We agree that such structures would facilitate understanding the role of these residues in function. To this end, we have made repeated efforts to determine the structure of hydrolyzed imipenem, cefotaxime or ampicillin in complex with the NDM-1 triple mutant. However, we have not succeeded in obtaining a structure with antibiotic present.

We have, however, **revised the paper** to better interpret how substitutions impact β -lactam hydrolysis in light of the recently published structures of NDM-1 in complex with imipenem by Feng et al. In this regard, Figure 6 has been re-done showing a structural alignment of the NDM-1 triple mutant and wild type NDM-1 (Fig. 6A) and a detailed illustration of the large changes of the loop 10 main chain and side chain orientations (Fig. 6B). The description of the loop 10 changes has also been revised in line 316 and lines 321-332. Further, a new Figure 7 shows the structure of the active site of NDM-1 in complex with ampicillin overlaid with the NDM-1 triple mutant structure (Fig. 7 A-C) and the structure of NDM-1 in complex with imipenem overlaid with the triple mutant structure (Fig. 7 D-F). A detailed view of interactions between the imipenem and NDM-1 is provided with a Ligplot in the newly added Figures S11. These structures show a different positioning of carbapenems in the active site versus structures with ampicillin and cefuroxime and we base a possible explanation for the stringent sequence requirements for NDM-1 for carbapenems versus penicillins and cephalosporins on these structures. This is described in paragraph 5 of the discussion (lines 395-404) of the revised paper. Finally, a more detailed description of the NDM mechanism is provided in lines 54-57 of the introduction.

Minor points:

1. *“While the introduction and results sections are clear and well-written, the discussion section would benefit from some revision. As indicated above, the discussion also has some notable omissions..”*

Response: See response to major point 1 above about the revision of the discussion.

2. *“Figure 1. Panel B. The mechanism appears to show an asparagine with a charged ammonium group – is this supposed to be the (uncharged) side chain amide? The structure in panel C would benefit from enlargement (or removal, given the overlap with Figure 4).”*

Response: The asparagine has been corrected in the revised Figure 1. Also, the structure in Fig. 1 panel C has been enlarged.

3. *“Figure 2 – Please add the stereochemistry for the beta-lactams. Please also correct the sidechain of imipenem, which should not have a carbon-carbon double bond”*

Response: The side chain of imipenem has been corrected and stereochemistry has been added to the revised Figure 2.

4. *“Figure 6 – His118 is listed twice in panel A”*

Response: Figure 6 has been revised as described above for the major points.

5. *“Table S1 appears to be missing from the Supporting Information”.*

Response: Table S1 (which is an Excel file) was included in the original submission as supplemental data. I am not sure why it was not available. Nevertheless, it has been submitted with the revised version.

6. *“Page 3 paragraph 4 – make clear the role of polar residues in binding the substrate carboxylate”*

Response: The paragraph has been revised to make clear the interaction with the substrate carboxylate (page 4, lines 60-62).

7. *“Please minimize the ‘deep sequencing’ jargon”*

Response: We have attempted to reduce sequencing jargon. For example, in paragraph 1 of the results section, “Illumina paired-end MiSeq sequencing” was changed to “next generation sequencing” (line 114). In paragraph 2 of results, “reads” was changed to “sequences” (line 118).

8. *“It would be of interest to define the methods looking at beta-lactams without side chains e.g.– clavamate would be of interest”*

Response: We were not able to obtain clavamate to test for this experiment.

9. *“Are the pH/buffer/temperature dependencies of the mutual enzymes the same as WT? (This information is essential at least for key mutants).”*

Response: The pH profile of the wild type and the K224R/G232A/N233Q triple mutant were investigated and found to be similar. This information has been added as the new Supplemental Figure 2 and described in the results section on page 11, lines 279-280.

10. “- Label N- and C-termini on crystal figures (4,6)”

Response: N- and C-termini are now labeled for Figures 4 and 6.

11. “Show stereoview electron density maps for active site residues.”

Response: Figure S4 has been added and shows a stereoview of the electron density map for the active site of the K224R/G232A/N233Q triple mutant structure.

Reviewer 2

Comments:

1. *The study represents an interesting analysis of the sequence requirements of NDM-1 to maintain activity for a broad range of β -lactam substrates. The take home message is that in their quest to design NDM-1 inhibitors researchers need to use carbapenem substrates for testing the effects of such compounds. In this context it is somewhat of a pity that the authors did not briefly elaborate on current advances in inhibitor design against MBLs in general, and NDM-1 in particular (see, for instance, recent studies from the Schofield and McGearry groups).*

Response: We agree that an important conclusion from the results is the need to use carbapenems in combination with new inhibitors. We have revised paragraph 2, lines 44-45 of the introduction to include mention of new inhibitor design and added relevant citations. Due to page/word count restrictions and the fact that inhibitor design, while important, is not the focus of the paper, we have not included an extensive discussion of inhibitors.

2. *Also, since MBLs from the B1 and B2 subgroups are related in an evolutionary sense, but the B2 representatives have a distinct preference for carbapenem substrates I wonder if any of the residues identified as important for the carbapenemase activity in NDM-1 are conserved in B2 MBLs? A more detailed comparison between NDM-1 and B2 MBLs may enhance the significance and broader impact of the study.*

Response: A new paragraph has been added to the discussion section (paragraph 5 of discussion, lines 376-392) comparing the results of the NDM study with the B2 enzyme, CphA, with respect to carbapenem hydrolysis and the narrow specificity of CphA compared to B1 enzymes like NDM. New Figure S7 shows the comparison using sequence logos. The comparison shows that many of the same active site residues are identified as important for both NDM-1 and CphA. Also, strikingly, for three positions, the wild-type residue is required for both enzymes but the identity of the residue is different for NDM-1 versus CphA. This observation indicates epistasis and suggests that these three positions may be important for

determining the narrow substrate specificity of CphA compared to NDM-1. Support for this hypothesis comes from previous studies showing that mutation of two of these positions in CphA broadens the specificity of the enzyme to include penicillins and cephalosporins. We thank the reviewer and reviewer 1 for this comment and believe this has broadened the scope of the paper.

3. Since MBLs from the B3 subgroup are evolutionarily distinct, but functionally quite similar to B1 MBLs an expansion of the discussion to include a sequence comparison between NDM-1 and at least a couple of well-known B3 MBLs (e.g. L1 and AIM-1) would possibly enhance the impact of the present study even further.

Response: As described in the response to point 2, we have added discussion of the comparison of the B1 NDM-1 versus the B2 CphA enzyme based on available codon randomization mutagenesis data. However, we have not added a B3 discussion in that there is not similar mutagenesis data for a B3 enzyme for comparison to the NDM-1 or CphA results.

4. The analysis of deep sequencing data is not an area I have great experience in, but as far as I can evaluate the study has been conducted in proficient manner. The presentation is equally proficient but the text may need some overhaul. Various grammatical errors pervade the manuscript, and the second paragraph in the Discussion section (page 12) appears rather speculative. On occasion (e.g. line 3 on page 12) residues are labeled in single character format (L121), while most of the time the three-letter abbreviation was used. Consistent use of abbreviations and notations should be ascertained throughout the manuscript.

Response: The manuscript has been edited to reduce grammatical errors and label amino acids with a consistent notation including L121. In response to the comment that paragraph 2 of the discussion is rather speculative, we have modified this paragraph to include discussion of imipenem chirality versus penicillins and cephalosporins and focus the paragraph on available structures of hydrolyzed imipenem and meropenem in complex with NDM-1. These structures show a different positioning of carbapenems in the active site versus structures with ampicillin and cefuroxime and base a possible explanation for the stringent sequence requirements for NDM-1 for carbapenems versus penicillins and cephalosporins. These changes are in paragraph 5 (lines 395-404) of the revised paper. In addition, a new Figure 7 has been added to more clearly show interactions between hydrolyzed imipenem and the NDM-1 enzyme and how changes in NDM-1 such as in the K224R/G232A/N233Q triple mutant affect imipenem but not ampicillin hydrolysis. The original Fig. 7 is now Fig. S6 of the revised paper. Finally, new Figure S11 has been added containing a Ligplot diagram illustrating the detailed contacts between imipenem and NDM-1 based on a recently published structure.

6. Lastly, is the term "hub residue" commonly used? This question may be a reflection of my ignorance but a brief definition when used first might be useful.

Response: "Hub residue" has been used in several publications including those cited when it was invoked and refers to residues that interact with other residues that are distant in the

primary amino acid sequence but close in the tertiary structure. However, a search of the literature reveals it is not often used. Therefore, we have eliminated the word “hub” and just state the residues serve to connect loop regions in the discussion section on page 14, lines 365-368.

Reviewer 3

Comments:

1. This manuscript reports a well conducted study related to antibiotic resistance. The hypothesis is sound and the question of interest, although the interest is quite specific to an expert audience. The work is of high quality, the methodology and analysis appear robust. The results of the deep-sequencing analysis comparing the effects of mutations on bacterial survival in the presence of 3 antibiotics is striking and is set in an appropriate light. The correlation between the results of deep sequencing and traditional sequencing appears sufficiently robust to support the conclusions and allows the reader to appreciate the advantage gained by applying that very high throughput strategy. A reasonable number and distribution of mutants were selected for more detailed analysis of the enzyme activity, and those results clearly support the plate screening results.

I disagree in part with the conclusions which I judge to be too general. The study was undertaken with 1 b-lactam antibiotic of each of 3 classes. The results therefore compare these 3 antibiotics. The conclusions state the demonstration of differences for the 3 classes. This was not shown. It can, however, be said that the results for the individual b-lactam antibiotics of each of 3 classes suggests that the observations may extend to the classes.

Part of the discussion illustrates this: ‘based on available structural information, carbapenems make fewer contacts with the enzyme than penicillins and cephalosporins (Fig. S4-S6)’; a single b-lactamase-bound inhibitor serves to illustrate the point. It should be toned down to reflect the difference between what is demonstrated or known for a single substrate, and for a class.

The authors addressed this in part (page 10): ‘To investigate whether this finding can be generalized to β -lactams in the same class, the triple mutant K224R/G232A/N233Q was tested for hydrolysis of another penicillin and carbapenem, i.e., benzylpenicillin and meropenem, respectively. As shown in Table S2, compared to wild-type NDM-1 enzyme, the triple mutant displays a 25-fold lower k_{cat}/K_M value for hydrolyzing benzylpenicillin but a 250-fold lower k_{cat}/K_M value for hydrolyzing meropenem. Therefore, the result is consistent with the idea that more extensive amino acid sequence information is required in the active site of NDM-1 for carbapenem hydrolysis compared to other b-lactam antibiotics.’ The tone here is acceptable: these results are consistent with the hypothesis of generality. Nonetheless, these are results only of one triple mutant and do not allow to conclude on the array of point mutants.

Response: The point that the conclusions drawn are too broad has been addressed at various places in the manuscript including the last paragraph of the introduction on p. 5, lines 97-100, p. 10, lines 243-244 and lines 254-255, and in the discussion on p. 13, lines 349-350, and lines 460-461. With these changes, we have attempted to adjust the tone based on the available results as suggested by the reviewer.

2. Page 9: amino acid substitutions at nonessential residue positions Phe64 and Asp225: to confirm that these positions can be substituted, it should also be demonstrated that their level of expression was similar to WT.

Response: Western blot data showing that expression levels for Phe64 and Asp225 mutants are similar to wild-type NDM-1 is now included in Figure S2 and noted in the results section on p. 10, lines 248-250.

3. Methods:

Calculation of effective number of substitutions (k^) is not clear because π is not defined.*

Response: The definition of π has been added to the explanation of the k^* calculation in Methods (page 21, lines 587-589).

4. What was the purity achieved for the different mutants that were purified and how was purity accounted for in determination of kinetic parameters (assuming it is not always very high, in the case of poorly expressed mutants for example) ? What is the level of background activity for hydrolysis of each of the substrates in absence of any NDM-1 mutant?

Response: All mutants that were used for enzyme kinetic studies were purified to >90% homogeneity as judged by SDS-PAGE. This has been added to the discussion of enzyme purification in Methods (lines 609-610). With regard to background activity, the hydrolysis of ampicillin, cefotaxime and imipenem in the absence of enzyme is not detectable over the time frame used for the initial velocity experiments performed to determine k_{cat}/K_M . This has been added to the enzyme kinetics section of Methods (p.22, lines 629-631).

Reviewers' Comments:

Reviewer #2:

Remarks to the Author:

The authors addressed many of the comments raised by the 3 reviewers. I let my colleagues evaluate the respective responses to their queries. I am largely satisfied with the response of the authors to my questions. The inclusion of the comparison between NDM-1 and CphA has certainly enhanced the breadth of the scope of the present study. It is thus somewhat unfortunate that the authors may not be aware of a site-saturation mutagenesis/in vitro evolution study that was recently published for the B3 MBL AIM-1 (see Hou et al Sci Rep 7 (2017): 40357). Keeping space limitations in mind there would be enough room to compare some of the active site mutations between NDM-1 and AIM-1. Since NDM-1, CphA and AIM-1 represent the 3 major group that make up the large family of MBLs associated with antibiotic resistance AIM-1 should really be included to widen the scope sufficiently to warrant publication in a high impact journal such as Nature Communications.

Reviewer #3:

Remarks to the Author:

All of the concerns I had noted were satisfactorily addressed by the authors. The additions and corrections that were made in response to the reviews add depth and clarify the message. Overall, this report of the molecular detail of (presumably) ancient and recent events in the evolution of MBL to hydrolyze different types of b-lactam antibiotics is skillful and convincing.

Response to reviewer comments

Responses to the reviewer comments are listed below. Note that the changes described in the responses are indicated by line numbers in the revised manuscript. The revisions in response to the reviews below are highlighted in cyan in the manuscript.

Reviewer 2

Comments:

Reviewer #2 (Remarks to the Author):

The authors addressed many of the comments raised by the 3 reviewers. I let my colleagues evaluate the respective responses to their queries. I am largely satisfied with the response of the authors to my questions. The inclusion of the comparison between NDM-1 and CphA has certainly enhanced the breadth of the scope of the present study. It is thus somewhat unfortunate that the authors may not be aware of a site-saturation mutagenesis/in vitro evolution study that was recently published for the B3 MBL AIM-1 (see Hou et al Sci Rep 7 (2017): 40357). Keeping space limitations in mind there would be enough room to compare some of the active site mutations between NDM-1 and AIM-1. Since NDM-1, CphA and AIM-1 represent the 3 major groups that make up the large family of MBLs associated with antibiotic resistance AIM-1 should really be included to widen the scope sufficiently to warrant publication in a high impact journal such as Nature Communications.

Response: A new paragraph has been added to the discussion section at the bottom of page 14 and top of page 15 describing the differences between subclass B1 enzymes such as NDM-1 with subclass B3 enzymes. In addition, a more detailed comparison is made between NDM-1 and the subclass B3 enzyme AIM-1. We have used the mutagenesis data from Hou et al indicated by reviewer 2 to compare positions that are in common in our mutagenesis study. We find that, although only 8 positions were studied in AIM-1, a similar trend exists in that the AIM-1 enzyme has less stringent sequence requirements for ampicillin hydrolysis versus imipenem. This finding suggests it may be a general property of subclass B1 and B3 enzymes that more sequence information in the active site is required to hydrolyze carbapenem versus other classes of β -lactam antibiotics. A new supplemental figure (Fig. S8) has been added to illustrate the NDM-1 active site in comparison to AIM-1.

Reviewer 3

Comments:

Reviewer #3 (Remarks to the Author):

All of the concerns I had noted were satisfactorily addressed by the authors. The additions and corrections that were made in response to the reviews add depth and clarify the message. Overall, this report of the molecular detail of (presumably) ancient and recent events in the evolution of MBL to hydrolyze different types of β -lactam antibiotics is skillful and convincing.

Response: We thank the reviewer for the comments.